# Subquadratic Overparameterization for Shallow Neural Networks

**Chaehwan Song**[1]*        **Ali Ramezani-Kebrya**[1]*

**Thomas Pethick**[1]     **Armin Eftekhari**[2]†     **Volkan Cevher**[1]

[1]Laboratory for Information and Inference Systems (LIONS), EPFL    [2]Umea University

ali.ramezani@epfl.ch

## Abstract

Overparameterization refers to the important phenomenon where the width of a neural network is chosen such that learning algorithms can provably attain zero loss in nonconvex training. The existing theory establishes such global convergence using various initialization strategies, training modifications, and width scalings. In particular, the state-of-the-art results require the width to scale quadratically with the number of training data under standard initialization strategies used in practice for best generalization performance. In contrast, the most recent results obtain linear scaling either with requiring initializations that lead to the "lazy-training", or training only a single layer. In this work, we provide an analytical framework that allows us to adopt standard initialization strategies, possibly avoid lazy training, and train all layers simultaneously in basic shallow neural networks while attaining a desirable subquadratic scaling on the network width. We achieve the desiderata via Polyak-Łojasiewicz condition, smoothness, and standard assumptions on data, and use tools from random matrix theory.

## 1 Introduction

Training a neural network involves solving a nonconvex optimization problem, which, in theory, might trap first-order methods such as gradient descent to fall in bad local minima or saddle points. However, empirical evidence suggests that first-order methods with random initialization can consistently find a global minimum, even with randomized labels [43]. Demystifying this observation is of central interest to deep learning.

Recently, a line of research [45, 4, 11, 29, 39, 12, 38] suggests that such an empirical success can possibly be explained by the *overparameterization* of neural networks, whose number of parameters exceeds the number of training data $n$. In particular, gradient descent converges linearly fast to a global optimum in a number of problems with models that have wide hidden layers [45, 12, 39].

Despite of these remarkable results, the natural key question *"How much should we overparameterize a neural network?"* remains open even for the toy example of two-layer neural networks. On one hand, it is widely accepted that, for two-layer neural networks, the number of parameters should grow linearly with $n$ (*e.g.,* [21, 38]). On the other hand, theoretical results either require much more parameters, or they are established under restrictive settings. Specifically,

---

*Equal contributions.

†This work was done while Armin Eftekhari was at EPFL.

35th Conference on Neural Information Processing Systems (NeurIPS 2021).

**Table 1:** Scaling with the number of training data in the overparameterization regime. QL=quadratic loss, CLL=convex and Lipschitz loss, SD=separable data.

| Depth | Algorithm | Setting | Activation | Scaling | Reference |
|-------|-----------|---------|------------|---------|-----------|
| 2 | GD on layer 1 | QL | ReLU | $\tilde{\Omega}(n^2)$ | Oymak and Soltanolkotabi [38] |
| $L$ | GD on layer $L$ | CLL | ReLU | $\tilde{\Omega}(n)$ | Kawaguchi and Huang [21] |
| 2 | GD | SD | ReLU | $\tilde{\Omega}(n^2)$ | Song and Yang [39] |
| 2 | GD | SD and QL | ReLU | $\tilde{\Omega}(n^6)$ | Du et al. [12] |
| $L$ | GD | SD and QL | ReLU | $\Omega(n^8 L^{12})$ | Zou and Gu [44] |
| 2 | GD | QL | Smooth | $\tilde{\Omega}(n^{\frac{3}{2}})$ | **This paper** |

- Kawaguchi and Huang [21] has proven the ideal $\tilde{\Omega}(n)$ scaling for deep neural networks. However, they apply gradient descent only to the last layer, which is not the case in practical scenarios.

- A similar issue exists in [39, 38], where the authors have shown that $\tilde{\Omega}(n^2)$ parameters suffice for two-layer neural networks, but only the first layers are trained. Furthermore, even with infinite width, Oymak and Soltanolkotabi [38] cannot guarantee zero training error with probability approaching to one.

The goal of this paper is to close the gap between theory and practice, without resorting to unrealistic assumptions such as those discussed above. We sharpen the results of Oymak and Soltanolkotabi [38] by proving that, with proper random initialization of each layer, training error approaches to zero with high probability, exponentially fast in the width of the network. In addition, we show that only $\tilde{\Omega}(n^{\frac{3}{2}})$ parameters suffice such that gradient descent converges to a global minimum with linear rate, which improves upon the state-of-the-art by a factor of $\tilde{O}(n^{\frac{1}{2}})$. We summarize the bounds on the number of parameters in terms of $n$ in Table 1.

While our analysis on gradient descent focuses on training error, it has been observed that overparameterization can lead to poor *generalization*. In particular, [7, 42, 15] have observed the phenomenon of *lazy training*. Chizat et al. [7] has explained lazy training as a model behaves similar to its linearization around the initialization. It is known that an overparameterized neural network is likely to be trapped in the lazy regime since the parameters will hardly vary over the course of training with gradient descent [12, 29, 45]. The same phenomenon has been observed for infinitely wide neural networks [19]. In this paper, we provide theoretical guidance to possibly avoid lazy training through proper initialization. Experimental results confirm that lazy training might be avoided with our theoretically inspired initialization so that the issues reported in [7] do not apply.

### 1.1 Summary of contributions

- We first focus on a general minimization problem assuming that the loss function satisfies Polyak-Łojasiewicz (PL) condition. We find sufficient conditions in terms of initialization for the convergence of gradient flow and gradient descent to a global minimum.

- We then focus on the special problem of training a two-layer neural network with quadratic loss and smooth activation, and show that $\tilde{\Omega}(n^{\frac{3}{2}})$ parameters are sufficient for gradient descent to converge to a global minimum with linear rate and probability approaching to one. We achieve *linear scaling* for the width when the number of input features is in $\tilde{\Omega}(\sqrt{n})$.

- We theoretically guide how to initialize the parameters of a neural network in the overparameterized regime of interest while possibly avoiding lazy training.

### 1.2 Further related work

In terms of techniques, our paper is closely related to [37, 38]. Similar to our Theorem 3, Oymak and Soltanolkotabi [38, Theorem 2.1] showed that gradient descent converges with linear rate when the Jacobian of the nonlinear mapping has smooth deviations, and the number of parameters grows quadratically with $n$. However, Oymak and Soltanolkotabi [38] assumed that gradient descent updates only the first layer. In this paper, we consider the case where gradient descent updates both layers simultaneously, and show that it suffices to have $\tilde{\Omega}(n^{\frac{3}{2}})$ parameters with a linear rate of convergence.

ReLU is an important instance of activation functions that does not satisfy the smoothness assumption. A line of research aims to relax this assumption by instead assuming the data is separable. For shallow neural networks, Du et al. [12] proved that gradient descent finds a global minimum if the width of the network scales $\tilde{\Omega}(n^6)$ assuming that no two data points are parallel. In a similar setting, Song and Yang [39] established convergence to a global minimum with the sufficient width of $\tilde{\Omega}(n^2)$. As a result, in the absence of the smoothness assumption, these papers require substantially more number of parameters to guarantee convergence to a global minimum.

The theoretical bounds for deep neural networks are even worse. For instance, Allen-Zhu et al. [1] required the total number of parameters of $\Omega(n^{24}L^{12})$ where $L$ is the number of layers. Zou and Gu [44] improved the scaling to $\Omega(n^8 L^{12})$. In our setting, *i.e.,* $L = 2$, these bounds become vacuous in most interesting regimes. Further, in [21], the authors showed that $\tilde{\Omega}(n)$ parameters is enough to achieve global convergence under the assumption that gradient descent updates only the last layer, which essentially reduces the problem to a simple least-squares regression.

Recently, Ji and Telgarsky [20], Chen et al. [5] showed that a polylogarithmic width suffices to achieve convergence for shallow and deep neural networks in an ergodic sense. We note that this is a weaker notion of convergence compared to the one we consider.

Li et al. [28] showed that gradient descent along with early stopping are robust to label noise on a constant fraction of labels in an overparameterized network. However, only the first layer is optimized in [28]. For possibly overparameterized and linear networks, Eftekhari [14] showed that gradient flow can successfully avoid lazy training assuming that the network has a layer with a single neuron. We note that our analysis does not require those restrictions.

Under an assumption similar to PL condition, Zou et al. [45] studied the problem of binary classification for a deep network with ReLU activation, which is a different problem compared to ours. In [40], the authors proved that gradient descent with overparameterization achieves zero-approximation when the underlying function that generates the labels has low-rank approximation. Their scaling requires perfect information about the target function, which is not the case in our paper. Under a variant of Xavier initialization, Daniely [9] found near optimal scaling for a binary classification problem trained by stochastic gradient descent. We note that the setting considered in our paper is more challenging than binary classification. Our results establish a new state-of-the-art on the required number of parameters in a nonrestrictive setting when both layers are trained at the same time. Recently, Nguyen and Mondelli [34] obtained subquadratic scaling for a deep neural network with pyramidal structure under an initialization that leads to lazy training. Our results do not have such restrictions.

Mean-field analysis was used to approximate a target distribution of parameters of a neural network by the empirical distributions [33, 32]. However, these results do not provide useful bounds on the scaling in terms of $n$, which is our focus in this paper.

Liu et al. [31] established global convergence when the function to minimize satisfies a variant of PL condition (local PL condition) assuming the map is Lipschitz continuous, which is not the case in our paper. Liu et al. [30] characterized the constancy of the neural tangent kernel via scaling properties of the norm of the Hessian matrix of the network. In this work, we focus on obtaining a sufficient number of parameters for gradient descent to converge to a global minimum with linear rate.

**Notation.** We use $\|\cdot\|$ to represent the Euclidean norm of a vector and Frobenius norm of a matrix. We use $\nabla$ to denote the Jacobian of a vector-valued and gradient of a scalar-valued function and $\nabla\Phi(a)\{b\}$ to represent the directional derivative of $\Phi$ along $b$. We use $\odot$ and $\otimes$ to denote the Hadamard (entry-wise) product and Kronecker product, respectively. For $A \in \mathbb{R}^{m \times n}$ and $t \in \mathbb{Z}_+$, we denote $A^{*t} \in \mathbb{R}^{m^t \times n}$ with its $a$-th column defined as $\mathrm{vec}(x_a \otimes \cdots \otimes x_a) \in \mathbb{R}^{m^t}$. We use lower-case bold font to denote vectors. Sets and scalars are represented by calligraphic and standard fonts, respectively. We use $[n]$ to denote $\{1, \cdots, n\}$ for an integer $n$. We use $\tilde{O}$ and $\tilde{\Omega}$ to hide logarithmic factors and use $\lesssim$ to ignore terms up to constant and logarithmic factors.

## 2 Problem, definitions, and assumptions

In this section, we set up a general compositional optimization problem. Then we focus on the special case of shallow neural networks in Section 5.

Let $\mathbf{w} \in \mathbb{R}^d$ denote a parameter vector where $d$ denotes the number of parameters. In a neural network, $\mathbf{w}$ consists of weights and biases of all layers. We consider the minimization problem

$$\min_{\mathbf{w} \in \mathbb{R}^d} h(\mathbf{w}) \tag{1}$$

where $h : \mathbb{R}^d \to \mathbb{R}_+$ is the composition of a loss function $f : \mathbb{R}^{\tilde{d}} \to \mathbb{R}_+$ and a nonlinear and nonconvex function $\Phi : \mathbb{R}^d \to \mathbb{R}^{\tilde{d}}$:

$$h(\mathbf{w}) = f(\Phi(\mathbf{w})) = f(\mathbf{z}) \tag{2}$$

where $\mathbf{z} = \Phi(\mathbf{w})$.

Before providing the details, let us highlight the simple idea behind the argument (see also [37]). Let $\mathbf{w}_0$ and $\overline{\mathbf{w}}$ denote the initial point and limit point when the gradient descent algorithm is run with some learning rate, respectively. The precise formulation of gradient descent is provided in Section 4. Let $\nabla\Phi^*(\overline{\mathbf{w}}) : \mathbb{R}^{\tilde{d}} \to \mathbb{R}^d$ denote the adjoint operator of $\nabla\Phi(\overline{\mathbf{w}})$. Since $\overline{\mathbf{w}}$ is a first-order stationary point of $h$, we have

$$0 = \nabla h(\overline{\mathbf{w}}) = \nabla\Phi^*(\overline{\mathbf{w}})\{\nabla f(\overline{\mathbf{z}})\}$$

where $\overline{\mathbf{z}} = \Phi(\overline{\mathbf{w}})$. Suppose that $\nabla\Phi^*(\overline{\mathbf{w}})$ is a nonsingular operator. Then $\nabla f(\overline{\mathbf{z}}) = 0$. If $\overline{\mathbf{z}}$ is a global minimizer of $f$, then $\overline{\mathbf{w}}$ is a global minimizer of $h$. To prove global convergence, it suffices to show that $\nabla\Phi^*$ is nonsingular within a neighborhood of the initialization $\mathbf{w}_0$, and that points reached by gradient descent remain within this neighborhood. We will prove that both statements hold with high probability for shallow neural networks.

We first define two notions that are useful to state a key lemma for our main results:

**Definition 1** (Near-isometry). *A linear mapping $T : \mathbb{R}^{d_1} \to \mathbb{R}^{d_2}$ is $(\mu, \nu)$-near-isometry if there exist $0 < \mu \leq \nu$ such that*

$$\mu \leq \sigma_{\min}(T) \leq \sigma_{\max}(T) \leq \nu. \tag{3}$$

**Definition 2** (Smoothness). *Let $\beta_\psi > 0$. A function $\psi : \mathbb{R}^{d_1} \to \mathbb{R}^{d_2}$ is $\beta_\psi$-smooth, if for all $\mathbf{u}, \mathbf{v} \in \mathbb{R}^{d_1}$, we have*

$$\sigma_{\max}(\nabla\psi(\mathbf{u}) - \nabla\psi(\mathbf{v})) \leq \beta_\psi \|\mathbf{u} - \mathbf{v}\|. \tag{4}$$

The following lemma shows that a smooth function, which is near-isometry at initialization, remains near-isometry for all nearby points of the initialization.

**Lemma 1.** *Suppose that $\Phi$ is $\beta_\Phi$-smooth and $\nabla\Phi^*(\mathbf{w}_0)$ is $(\mu_\Phi, \nu_\Phi)$-near-isometry. Then, for all $\mathbf{w} \in \mathrm{ball}(\mathbf{w}_0, \rho_\Phi)$, we have*

$$\frac{\mu_\Phi}{2} \leq \sigma_{\min}(\nabla\Phi^*(\mathbf{w})) \leq \sigma_{\max}(\nabla\Phi^*(\mathbf{w})) \leq \frac{3\nu_\Phi}{2} \tag{5}$$

*where*

$$\rho_\Phi = \frac{\mu_\Phi}{2\beta_\Phi}. \tag{6}$$

Intuitively, if $\nabla\Phi^*(\mathbf{w}_0)$ is a $(\mu_\Phi, \nu_\Phi)$-near-isometry, then one would expect $\nabla\Phi^*$ to remain near-isometry for all nearby points.

**Definition 3** (PL condition [3]). *A function $\psi : \mathbb{R}^{d_1} \to \mathbb{R}$ satisfies the PL condition if there exists $\alpha_\psi > 0$ such that, for all $\mathbf{u} \in \mathbb{R}^{d_1}$, we have*

$$\psi(\mathbf{u}) \leq \frac{\|\nabla\psi(\mathbf{u})\|^2}{2\alpha_\psi}. \tag{7}$$

We note that strongly convex functions satisfy a minor variation of the PL condition in (7).

In our analysis, we will assume that $\Phi$ and $f$ satisfy the following properties:

**Assumption 1** (Basic assumptions for $\Phi, f$).

- $\Phi$ *is twice-differentiable and* $\beta_\Phi$*-smooth.*

- $f$ *is twice-differentiable, satisfies the PL condition with* $\alpha_f$*, and* $\min f(\mathbf{z}) = 0$*.*

Despite $f$ satisfies the PL condition, the nonconvex $\Phi$ can render $h$ nonconvex, and hence difficult to minimize in theory. However, we show that fast convergence of gradient descent to a global minimum can be established with appropriate initialization.

The intuition behind these assumptions is that to achieve nonsingularity of $\nabla\Phi^*$, we approximate $\nabla\Phi^*(\mathbf{w}_0)$ at initialization and bound $\nabla\Phi^*(\mathbf{w}_0) - \nabla\Phi^*(\mathbf{w}_i)$ at iteration $i$ using the fact that $\|\mathbf{w}_0 - \mathbf{w}_i\|$ is sufficiently small by the overparameterization. In the special case of shallow neural networks, we expect a similar argument applies even when the activation function is ReLU. Adapting our analysis for such extensions is an interesting area of future work.

# 3 Gradient flow

In this section, we consider gradient flow, which can be viewed as the limit of gradient descent for infinitesimally small learning rates. Inspired by the analysis of gradient flow, we provide an upper bound on the length of the trajectory traversed by gradient descent iterates and then find a sufficient condition in terms of initialization to establish its convergence to a global minimum. We focus on gradient descent in Section 4.

Let $t \geq 0$ and consider the gradient flow, which is initialized at $\mathbf{w}_0 \in \mathbb{R}^d$ and traverses the curve $\gamma : \mathbb{R}_+ \to \mathbb{R}^d$, given by

$$\dot{\gamma}(t) = \frac{\mathrm{d}\gamma(t)}{\mathrm{d}t} = -\nabla h(\gamma(t)) \tag{8}$$

where $\gamma(0) = \mathbf{w}_0$.

We now calculate the length of the curve $\gamma$. Suppose that $\nabla\Phi^*(\mathbf{w}_0)$ is $(\mu_\Phi, \nu_\Phi)$-near-isometry. Using Lemma 1, in the following lemma, we control the length inside of $\mathrm{ball}\,(\mathbf{w}_0, \rho_\Phi)$. See Appendix B for the proof.

**Lemma 2.** *Let* $t \geq 0$ *and let* $\ell(t)$ *denote the length of the curve* $\gamma$ *in (8), restricted to the interval* $[0, t]$*. Let* $t_\Phi \in (0, \infty]$ *be the smallest value such that* $\gamma(t_\Phi) \notin \mathrm{ball}(\mathbf{w}_0, \rho_\Phi)$*. Suppose* $\nabla\Phi^*(\mathbf{w}_0)$ *is* $(\mu_\Phi, \nu_\Phi)$*-near-isometry. Then, for all* $t \leq t_\Phi$*, we have*

$$\ell(t) = O\left(\frac{\nu_\Phi\sqrt{h(\mathbf{w}_0)}}{\mu_\Phi^2\sqrt{\alpha_f}}\right).$$

Lemma 2 implies that if the objective value at initialization, $h(\mathbf{w}_0)$, is sufficiently small, then we can localize gradient flows to a region around $\mathbf{w}_0$. Combining with Lemma 1, we show that the limit point of gradient flow is a global minimum. This theorem is formally stated below.

**Theorem 1** (Gradient flow). *Let* $\mathbf{w}_0 \in \mathbb{R}^d$*. Suppose that* $\Phi$ *and* $f$ *satisfy Assumption 1 and* $\nabla\Phi^*(\mathbf{w}_0)$ *is* $(\mu_\Phi, \nu_\Phi)$*-near-isometry. If* $\mathbf{w}_0$ *satisfies*

$$h(\mathbf{w}_0) = O\left(\frac{\alpha_f\mu_\Phi^6}{\beta_\Phi^2\nu_\Phi^2}\right), \tag{9}$$

*then the gradient flow* $\gamma$ *in (8) converges to a global minimum.*

*Proof of Theorem 1.* Proper initialization in (9) ensures $\ell(t_\Phi) < \rho_\Phi$, which implies that

$$\|\gamma(t_\Phi) - \mathbf{w}_0\| = \|\gamma(t_\Phi) - \gamma(0)\| < \rho_\Phi. \tag{10}$$

Therefore, $\gamma(t) \in \mathrm{ball}(\mathbf{w}_0, \rho_\Phi)$ for all $t \geq 0$, and the length of $\gamma$ is upper bounded by $\rho_\Phi$ using Lemma 2. Hence, the gradient flow $\gamma$ converges, *i.e.,* the limit point $\overline{\mathbf{w}} \in \mathbb{R}^d$ exists and satisfies

$$\|\overline{\mathbf{w}} - \mathbf{w}_0\| \leq \rho_\Phi. \tag{11}$$

Combining (5) and (11), we have

$$\frac{\mu_\Phi}{2} \leq \sigma_{\min}(\nabla\Phi^*(\overline{\mathbf{w}})) \leq \sigma_{\max}(\nabla\Phi^*(\overline{\mathbf{w}})) \leq \frac{3\nu_\Phi}{2}.$$

In particular, we note that $\nabla\Phi^*(\overline{\mathbf{w}})$ is nonsingular. So we have $\nabla f(\overline{\mathbf{z}}) = 0$. Since $f$ satisfies the PL condition in (7), $\overline{\mathbf{z}}$ is a global minimizer of $f$, and $\overline{\mathbf{w}}$ is a global minimizer of $h$ in (1). $\qquad\square$

# 4  Gradient descent

We now view gradient descent as the discretization of gradient flow, and show that a similar argument as in Section 3 holds for gradient descent.

Let $\eta > 0$ denote the learning rate and let $i \geq 0$. The gradient descent update rule is given by

$$\mathbf{w}_{i+1} = \mathbf{w}_i - \eta \nabla h(\mathbf{w}_i). \tag{12}$$

To study gradient descent, in addition to the previous assumptions on $\Phi$ and $f$ for the case of gradient flow described in Theorem 1, we also assume that $f$ is smooth, *i.e.,* there exists $\beta_f \geq 0$ such that, for all $\mathbf{z}, \mathbf{z}' \in \mathbb{R}^{\tilde{d}}$, we have

$$f(\mathbf{z}) - f(\mathbf{z}') \leq \langle \mathbf{z} - \mathbf{z}', \nabla f(\mathbf{z}') \rangle + \frac{\beta_f}{2} \|\mathbf{z} - \mathbf{z}'\|^2.$$

Smoothness of $f$ allows safe discretization of gradient flow without deviating too much from its trajectory. The following result is the analogue of Theorem 1 for gradient descent; see Appendix C for the proof.

**Theorem 2** (Gradient descent). *Let $\mathbf{w}_0 \in \mathbb{R}^d$. Suppose that $\Phi$ and $f$ satisfy Assumption 1, $f$ is $\beta_f$-smooth, and $\nabla \Phi^*(\mathbf{w}_0)$ is $(\mu_\Phi, \nu_\Phi)$-near-isometry. Suppose that gradient descent is executed with sufficiently small learning rate*

$$\eta = O\left( \frac{1}{\beta_\Phi \|\nabla f(\Phi(\mathbf{w}_0)\| + \beta_f \mu_\Phi^2 + \beta_f \nu_\Phi^2} \right), \tag{13}$$

*and $\mathbf{w}_0$ satisfies* (9).

*Then the sequence of iterates $\{\mathbf{w}_i\}_{i \geq 0}$ converges to a global minimum of $h$ exponentially fast.*

*In addition, the rate of convergence is given by*

$$h(\mathbf{w}_i) \leq (1 - C\eta\alpha_f \mu_\Phi^2)^i h(\mathbf{w}_0) \tag{14}$$

*where $C$ is a universal constant.*

To prove Theorem 2, we first compute the length of the trajectory traversed by gradient descent iterates. We then use the smoothness of $f$ and follow the descent inequality to lower bound $f(\mathbf{z}_i) - f(\mathbf{z}_{i+1})$. Finally, we compute the local Lipschitz constant of $f$.

**Remark 1.** *The idea of initializing a nonconvex problem close to a global minimum has a long history in nonconvex optimization, particularly in matrix factorization; see [6] and references therein. The observation that the length of the learning trajectory is short in the overparameterization regime has a precedent in [12, 37]. From an algorithmic perspective, the idea of linearizing $\Phi$ when minimizing $h = f \circ \Phi$ is studied in nonlinear regression and the Gauss-Newton method [35].*

In order to apply Theorem 2, the key step is to verify that $h(\mathbf{w}_0)$ satisfies (9). In Section 5, we focus on the special case of shallow neural networks and improve the state of the art.

# 5  Shallow neural networks

In this section, we consider the problem of training shallow neural networks with gradient descent. Our strategy is to cast this problem as a special case of problem (1) and then apply Theorem 2 to establish global convergence. We start with the formal problem statement.

## 5.1  Setup, assumptions, and initialization

Consider a shallow neural network with $d_0$ inputs, one hidden layer that consists of $d_1$ hidden nodes, and $d_2$ outputs. This shallow network is specified by the map

$$\begin{aligned} \mathbb{R}^{d_0} &\mapsto \mathbb{R}^{d_2} \\ \mathbf{x} &\mapsto V \cdot \phi(W\mathbf{x}), \end{aligned} \tag{15}$$

where $W \in \mathbb{R}^{d_1 \times d_0}$, $V \in \mathbb{R}^{d_2 \times d_1}$, and $\phi : \mathbb{R} \to \mathbb{R}$ is an activation function, which is applied entry-wise. Let $\mathbf{x}_i \in \mathbb{R}^{d_0}$ and $y_i \in \mathbb{R}^{d_2}$ denote the $i$-th training data and label, respectively, for $i \in [n]$. By concatenating the training data and their labels, we form the matrices $X \in \mathbb{R}^{d_0 \times n}$ and $Y \in \mathbb{R}^{d_2 \times n}$. Let denote $\Theta = (W, V) \in \mathbb{R}^{d_1 \times d_0} \times \mathbb{R}^{d_2 \times d_1}$ and $Z = \Phi(\Theta) = V \cdot \phi(WX) \in \mathbb{R}^{d_2 \times n}$. The fitting problem can be cast as (1) where

$$h(\Theta) = f(\Phi(\Theta)) = \|V\phi(WX) - Y\|^2. \tag{16}$$

**Remark 2.** *We assume that the activation function $\phi : \mathbb{R} \to \mathbb{R}$ is twice-differentiable. Despite this assumption excludes the popular ReLU, it is still possible to apply our results to smooth approximations of ReLU such as the softplus or Gaussian error Linear Units (GeLU) [18, 34]. We note that softplus [13] or GeLU [10] often achieve similar or superior performance compared to the ReLU [8, 16, 23, 22, 41].*

**Definition 4** (Hermite norm [36])**.** *Let $\phi : \mathbb{R} \to \mathbb{R}$. The Hermite norm of $\phi$ is given by $\|\phi\|_{\mathcal{H}} = \sqrt{\sum_{i=0}^{\infty} c_i^2}$ where $c_i$ denotes the $i$-th Hermite coefficients of $\phi$ given by:*

$$c_i = \langle \phi, q_i \rangle_{\mathcal{H}} = \frac{1}{\sqrt{2\pi}} \int \phi(x) q_i(x) \exp\left(-\frac{x^2}{2}\right) \mathrm{d}x$$

*and $q_i : \mathbb{R} \to \mathbb{R}$ is the $i$-th Hermite polynomial for $i \geq 0$.*

In this section, we assume that $\phi$, $f$, and data satisfy the following properties:

**Assumption 2** (Assumptions for shallow neural networks)**.**

- $\phi$ is twice-differentiable, $\phi(0) = 0$, $\sup_a |\dot{\phi}(a)| = \dot{\phi}_{\max} < \infty$, $\sup_a |\ddot{\phi}(a)| = \ddot{\phi}_{\max} < \infty$, and $\|\phi\|_{\mathcal{H}} < \infty$. The loss function $f$ is quadratic (16).

- $\|\mathbf{x}_i\| = 1$, $\|Y\| \leq 1$, and $\sigma_{\max}(V_k) = O\left(\frac{\dot{\phi}_{\max}}{\ddot{\phi}_{\max}}\right)$ for $i \in [n]$ and $k \geq 0$.

The assumption on $\phi$ hold for GeLU, sigmoid, and tanh. The assumption $\phi(0) = 0$ is to simplify the derivations and we suspect that it can be removed at the expense of more complicated expressions. The bounded Hermite norm is a mild assumption, which is used to obtain an upper bound on $\sigma_{\max}(\phi(W^0 X))$ in terms of the Hermite coefficients of $\phi$. See Appendix E.1 for details. The assumption on the data is fairly mild and standard in the overparameterization literature as we can always normalize the data [29, 20]. Similar boundedness assumptions to the last assumption are commonly used in nonconvex optimization to guarantee convergence [24]. Moreover, such a bound naturally holds by applying a projection step to the gradient descent update rule, which we plan to adopt as a future work.

**Initialization.** We first consider the initialization scheme:

$$W_0 \sim \mathcal{N}(0, \omega_1^2), \quad V_0 \sim \mathcal{N}\left(0, \omega_2^2\right). \tag{17}$$

In Section 6, we study the implications of our initialization and show how to possibly avoid lazy training by varying $(\omega_1, \omega_2)$.

## 5.2 Main results for shallow neural networks

For shallow networks as described above, we verify in Appendix D that the key conditions in Lemma 1 hold with high probability. Combining with Theorem 2, we establish the global convergence guarantees. The proof in Appendix E uses standard tools from random matrix theory to control the random variables involved with initialization. We first estimate variables $\mu_\Phi, \nu_\Phi$ defined in Definition 1 and $\beta_\Phi$ in (4) for the neural network described in Section 5.1.

**Lemma 3** (Estimation of $\mu_\Phi, \nu_\Phi, \beta_\Phi$)**.** *Suppose that a shallow neural network, which is constructed in Section 5.1, satisfies Assumption 2. Then we have*

$$
\begin{aligned}
\mu_\Phi &= \sigma_{\min}(\phi(W_0 X)), \\
\nu_\Phi &= \dot{\phi}_{\max} \sigma_{\max}(X) \sigma_{\max}(V_0) + \sigma_{\max}(\phi(W_0 X)), \\
\beta_\Phi &= \sqrt{2} \sigma_{\max}(X) \left(\dot{\phi}_{\max} + \ddot{\phi}_{\max} \chi_{\max}\right)
\end{aligned}
\tag{18}
$$

*where $\chi_{\max} = \sup_V \sigma_{\max}(V)$.*

**Remark 3.** *The terms $\sigma_{\min}(\phi(W_0 X))$ and $\sigma_{\max}(\phi(W_0 X))$ in (18) play a critical role in our analysis. In [12, 39], strictly positiveness of the eigenvalues of Gram matrix is the primary tool to show the convergence. Oymak and Soltanolkotabi [38] also followed a similar argument using the neural network covariance matrix. The underlying intuition seems similar to Lemma 3. However, the resulting bounds are different since gradient descent updates $(W, V)$ simultaneously in our problem setup, which is more realistic.*

By combining Lemma 3 and the results on global convergence of gradient descent in Section 4, we establish global convergence for shallow neural network.

**Theorem 3** (Shallow network with gradient descent). *Consider the shallow network described in Section 5.1 that satisfies Assumption 2 and $\tau^{r_1}|\phi(a)| \leq |\phi(\tau a)| \leq \tau^{r_2}|\phi(a)|$ for all $a$, $0 < \tau < 1$, and some constants $r_1, r_2$.[3] Suppose that $\Theta_0$ is randomly initialized as in (17) with $\omega_1$ and $\omega_2$, which satisfy*

$$\omega_1 \omega_2 \lesssim \frac{1}{\sqrt{d_0 d_1}}, \tag{19}$$

*and suppose that the hidden layer width $d_1$ satisfies*

$$d_1 = \tilde{\Omega}\left(\xi(\mathcal{C}_\delta, t, \phi, \{c_i\}_{i \geq 0}) \frac{\sigma_{\max}(X)^2 \sqrt{n}}{\sigma_{\min}(X^{*t})^3}\right) \tag{20}$$

*where $\mathcal{C}_\delta$ is a set of constants, $\xi$ is a term independent to $d_0, n$, $t$ is a constant such that $n \simeq d_0^t$, and $X^{*t} \in \mathbb{R}^{d_0^t \times n}$ is derived from Khatri-Rao product with its $a$-th column defined as $\text{vec}(x_a \otimes \cdots \otimes x_a) \in \mathbb{R}^{d_0^t}$. Then gradient descent converges to a global minimum exponentially fast with probability at least $1 - \psi(\phi, \xi, d_0, d_1, d_2, X)$.[4] See Appendix E.6 for the exact expressions of $\xi$ and $\psi$.*

**Remark 4.** *Theorem 3 shows that, with sufficient degree of overparameterization, gradient descent finds a global minimum, except with an arbitrary small probability. Note that we need two conditions for Theorem 3 to hold, both of which are related to the overparameterization of the network. The condition (19) is for the concentration of random matrices, to make $\psi$ arbitrary small, and (20) is for the locality of gradient descent.*

## 5.3 Order analysis

We first decompose the random matrix $\phi(X^\top W_0^\top)\phi(W_0 X)$ into independent random matrices. We then apply concentration inequalities to establish an upper bound on $\sigma_{\max}(\phi(W_0 X))$ and a lower bound on $\sigma_{\min}(\phi(W_0 X))$ through the Hermite decomposition of $\phi(W_0 X)$ and note that with high probability,

$$\sqrt{\frac{c_t^2}{t!} d_1} \sigma_{\min}(X^{*t}) \lesssim \sigma_{\min}(\phi(W_0 X)) \lesssim \sigma_{\max}(\phi(W_0 X)) \lesssim \sqrt{c_0^2 dn}.$$

We also find an upper bound on $h(\Theta_0)$ at initialization. Substituting $\nu_\Phi, \mu_\Phi, \beta_\Phi$ into (9), we obtain the sufficient condition in (20). We note that $\xi(\mathcal{C}_\delta, t, \phi, \{c_i\}_{i \geq 0})$ can be viewed as a constant w.r.t. $d_0$, $d_1$, and $n$. For $t = 1$, it requires $n \simeq d_0$, which is not a common setting in practice. For $t \geq 2$, we suppose that $n \simeq d_0^t$, which is the case in practice and estimate $\sigma_{\max}(X) \simeq \sqrt{\frac{n}{d_0}}$ and $\sigma_{\min}(X^{*t}) \simeq \sqrt{\frac{n}{d_0^t}} \simeq 1$ along the lines of [38, Section 2.1]. Substituting $\sigma_{\max}(X)$ and $\sigma_{\min}(X^{*t})$ into (20), we have

$$d_1 \gtrsim \frac{n^{\frac{3}{2}}}{d_0}. \tag{21}$$

Therefore, the overall overparameterization degree becomes $d_0 d_1 \simeq \tilde{\Omega}(n^{\frac{3}{2}})$, which is sufficient for gradient descent to find a global minimum at a linear rate except with an arbitrary small probability. We note that an optimal linear scaling for the width $d_1 \simeq \tilde{O}(n)$ is sufficient when the number of input features is sufficiently large $d_0 \simeq \tilde{\Omega}(\sqrt{n})$, which improves upon the results of [38] by a factor of $\tilde{O}(n^{\frac{1}{2}})$. Furthermore, unlike [38], we adopt standard initialization strategies in Theorem 3.

---

[3]The last assumption holds for popular activation functions such as sigmoid, tanh, and ELU, and can be relaxed if $\omega_1 = 1$ in (17).

[4]$\psi$ can be arbitrary small.

# 6 Lazy training and experimental evaluation

Following the theoretically motivated initialization in Theorem 3, we set $\omega_1 \omega_2 \simeq \frac{1}{\sqrt{d_0 d_1}}$. This gives rise to a broad family of initialization schemes as one varies the ratio $\omega_2/\omega_1$. Interestingly, we note that popular initialization schemes such as LeCun [26] and He initialization [17] belong to this family. The purpose of this section is to empirically investigate the impact the choice of this ratio has on generalization of shallow networks.

To this end, we will look at the generalization error of varying initializations in the more practical setting of stochastic gradient descent (SGD). Specifically, we fix the product of the weight initialization $\omega_1 \omega_2$ and then proceed by varying

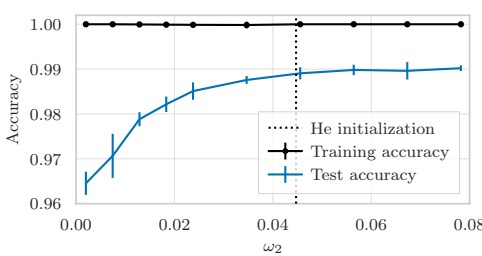

**Figure 1:** *Training and test error on MNIST for different $\omega_2$. Error bars indicates the 95% confidence interval computed over 5 independent runs. The setup details are provided in Appendix G.*

$\omega_2$. To ensure that perfect generalization is possible, we adopt the teacher-student setup, where, for the teacher network, we train a two-layer fully connected neural network, on MNIST [25] until SGD reaches zero training error. The student networks are trained for 300 epochs to ensure convergence. The results are shown in Figure 1. We use mean-square loss and a smooth activation function (GeLU [18]) for the student network to match the problem setup as closely as possible.

In Figure 1, we observe that while SGD achieves zero training error for every $\omega_2$, as suggested by Theorem 3 applicable in the full batch setting, the generalization ability increases as the ratio $\omega_2/\omega_1$ grows. It is also interesting to observe that the popular He initialization scheme corresponds to a rather balanced ratio that lies at the boundary of the well-performing region. In our experiments, we used He initialization to fix the value $\omega_1 \omega_2$. This tendency suggests that a wide family of initialization schemes could generalize well as long as the ratio $\omega_2/\omega_1$ is not too small.

**Comment on lazy training.** It is important to address the so called lazy regime when generalization is of concern. Let

$$\tilde{h}(\Theta) := h(\Theta_0) + \langle \nabla h(\Theta_0), \Theta - \Theta_0 \rangle$$

be the linearized function of $h$ around $\Theta_0$ and let $\Theta_i$ and $\tilde{\Theta}_i$ denote the iterates of gradient descent at time $i$. The lazy training regime refers to the case where the training trajectory stays close to this linearization, i.e. $\|h(\Theta_i) - \tilde{h}(\tilde{\Theta}_i)\| \simeq 0$ for all $i$ [7]. Such a linearization occurs in infinitely wide neural networks [7], which have been shown to generalize well in some settings [2, 27]. However, in our case of subquadratic (finite) width, the lazy regime might lead to poor generalization. To gain insight on when we cannot avoid it with certainty, let us make a simple rewriting of our network assuming $\phi$ is homogeneous:

$$\Phi(\Theta) = \alpha V \phi(WX)$$

with $V_0 \sim \mathcal{N}(0, 1)$ and $W_0 \sim \mathcal{N}(0, 1)$ where the standard deviations are pulled out as a scaling factor $\alpha = \omega_1 \omega_2 \simeq 1/\sqrt{d_0 d_1}$. This seems to fit into an example in [7, Appendix A.2] suggesting lazy training as $d_1 \to \infty$. However, their results require an odd activation function and infinite width, while our activation function is required not to be odd (see the proof in Appendix E) and our results are under subquadratic (finite) width. Instead, to study lazy training, we explicitly compute an upper bound on $\|h(\Theta_i) - \tilde{h}(\tilde{\Theta}_i)\|$ in Appendix F following [7, Theorem 2.3].

It turns out that the upper bound becomes $\infty$ when $\omega_1 \ll \omega_2$, and it becomes zero when $\omega_1 \gg \omega_2$. Our analysis suggests that shallow neural networks can avoid lazy training provided that $\omega_2/\omega_1 \to \infty$. This analysis is corroborated by the empirical results showing that the generalization capability improves as $\omega_2$ grows in Figure 1. On the other hand, if $\omega_2/\omega_1 \to 0$, then lazy training is bound to happen asymptotically. For details, see Appendix F. Finally, we note that we have not theoretically claimed that our initialization is guaranteed to be non-lazy, since doing so would require establishing a lower bound on $\|h(\Theta_i) - \tilde{h}(\tilde{\Theta}_i)\|$, which is an interesting problem for future work. Instead, our discussion above only provides a necessary condition for non-lazy training, and a sufficient condition for lazy training.

# 7 Conclusions and future work

In this paper, we prove the linear convergence of first-order methods on subquadratically overparameterized two-layer neural networks with smooth activation functions. Our theoretical analysis is compatible with standard initialization strategies, which can potentially avoid lazy training. We train both layers simultaneously and achieve a desirable subquadratic scaling on the width of the network. In particular, we note that a linear scaling for the width $d_1 \simeq \tilde{O}(n)$ is sufficient when the number of input features is sufficiently large $d_0 \simeq \tilde{\Omega}(\sqrt{n})$. We use tools from random matrix theory under standard assumptions on data and leverage on the assumption that the loss satisfies Polyak-Łojasiewicz condition. We carefully find an explicit upper bound and lower bound on singular values of the outputs of the first layer at initialization with high probability under general initialization.

It is natural to ask whether we can attain similar degree of overparameterization with nonsmooth activation functions such as ReLU. We plan to adapt our analysis for such extensions as a future work. While our analysis provides a necessary condition for avoiding lazy training, it is interesting to develop sufficient conditions in the future. In particular, developing lower bounds on $\|h(\Theta_i) - \tilde{h}(\tilde{\Theta}_i)\|$ will be a key to fully characterize lazy training.

Finally, as a theoretical work, we do not anticipate any potential negative societal impacts of our paper. However, the long-term impacts of our work may depend on how machine learning algorithms are used in society.

## Acknowledgments and Disclosure of Funding

The authors would like to thank Fabian Latorre, Fanghui Liu, and Paul Rolland for helpful discussions.

This project has received funding from the European Research Council (ERC) under the European Union's Horizon 2020 research and innovation programme (grant agreement n° 725594 - time-data). This project was sponsored by the Department of the Navy, Office of Naval Research (ONR) under a grant number N62909-17-1-2111. This work was supported by Hasler Foundation Program: Cyber Human Systems (project number 16066). Research was sponsored by the Army Research Office and was accomplished under Grant Number W911NF-19-1-0404.

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
