# A  Proof of Lemma 1

Intuitively, if $\nabla\Phi^*(\mathbf{w}_0)$ is a $(\mu_\Phi, \nu_\Phi)$-near-isometry, then one would expect $\nabla\Phi^*$ to remain near-isometry for all nearby points. Formally, let $A, B \in R^{m \times n}$ and let singular values of a matrix are ordered such that $\sigma_i(A) \geq \sigma_j(A)$ and $\sigma_i(B) \geq \sigma_j(B)$ for $1 \leq i \leq j \leq \min\{m, n\}$. Using Weyl's inequality and for $i + j - 1 \leq \min\{m, n\}$, we have:

$$\sigma_{i+j-1}(A + B) \leq \sigma_i(A) + \sigma_j(B). \tag{A.1}$$

More formally, suppose that $\mathbf{w} \in \mathbb{R}^d$ satisfies

$$\|\mathbf{w} - \mathbf{w}_0\| \leq \frac{\mu_\Phi}{2\beta_\Phi} = \rho_\Phi. \tag{A.2}$$

If $\nabla\Phi^*(\mathbf{w}_0)$ is $(\mu_\Phi, \nu_\Phi)$-isometry in the sense of Definition 1, then applying Weyl's inequality (A.1) along with using smoothness and (A.2), we have

$$\begin{aligned}
\sigma_{\min}(\nabla\Phi^*(\mathbf{w})) &\geq \sigma_{\min}(\nabla\Phi^*(\mathbf{w}_0)) - \sigma_{\max}(\nabla\Phi^*(\mathbf{w}) - \nabla\Phi^*(\mathbf{w}_0)) \\
&\geq \mu_\Phi - \beta_\Phi \|\mathbf{w} - \mathbf{w}_0\| \\
&\geq \frac{\mu_\Phi}{2}.
\end{aligned}$$

Using a similar argument, we establish an upper bound $\sigma_{\max}(\nabla\Phi^*(\mathbf{w}))$:

$$\sigma_{\max}(\nabla\Phi^*(\mathbf{w})) \leq \sigma_{\max}(\nabla\Phi^*(\mathbf{w}_0)) + \sigma_{\max}(\nabla\Phi^*(\mathbf{w}) - \nabla\Phi^*(\mathbf{w}_0)) \leq \nu_\Phi + \frac{\mu_\Phi}{2} \leq \frac{3\nu_\Phi}{2}.$$

# B  Proof of Lemma 2

Let $t \geq 0$ and denote

$$\zeta(t) = \Phi(\gamma(t)) \tag{A.3}$$

so we have

$$h(\gamma(t)) = f(\Phi(\gamma(t)) = f(\zeta(t)). \tag{A.4}$$

Taking the first-order derivative w.r.t. $t$, we have

$$\begin{aligned}
\dot{\zeta}(t) &= \nabla\Phi(\gamma(t))\{\dot{\gamma}(t)\} \\
&= -\nabla\Phi(\gamma(t))\{\nabla h(\gamma(t))\}.
\end{aligned} \tag{A.5}$$

Note that we have

$$\begin{aligned}
\frac{\mathrm{d}h(\gamma(t))}{\mathrm{d}t} &= \nabla h(\gamma(t))\{\dot{\gamma}(t)\} \\
&= -\nabla h(\gamma(t))\{\nabla h(\gamma(t))\} \\
&= -\|\nabla h(\gamma(t))\|^2.
\end{aligned} \tag{A.6}$$

Length of the segment of the curve $\gamma_K$ restricted to the interval $[0, t]$ is given by

$$
\begin{aligned}
\ell(t) &= \int_0^t \|\dot{\gamma}(\tau)\| \, \mathrm{d}\tau \\
&= \int_0^t \|\nabla h(\gamma(\tau))\| \, \mathrm{d}\tau \\
&\leq \int_0^t \sigma_{\max}(\nabla \Phi^*(\gamma(\tau)) \cdot \|\nabla f(\zeta(\tau))\| \, \mathrm{d}\tau \\
&\lesssim \nu_\Phi \int_0^t \|\nabla f(\zeta(\tau))\| \, \mathrm{d}\tau.
\end{aligned}
\tag{A.7}
$$

To control the norm in the last line of (A.7), we note that

$$
\begin{aligned}
-\frac{\mathrm{d}\sqrt{f(\zeta(\tau)) - f(\zeta(t))}}{\mathrm{d}\tau} &= -\frac{\frac{\mathrm{d}f(\zeta(\tau))}{\mathrm{d}\tau}}{2\sqrt{f(\zeta(\tau)) - f(\zeta(t))}} \\
&= -\frac{\langle \nabla f(\zeta(\tau)), \dot{\zeta}(\tau) \rangle}{2\sqrt{f(\zeta(\tau)) - f(\zeta(t))}} \\
&= \frac{\langle \nabla f(\zeta(\tau)), \nabla \Phi(\gamma(\tau)) \{\nabla h(\gamma(\tau))\} \rangle}{2\sqrt{f(\zeta(\tau)) - f(\zeta(t))}} \\
&= \frac{\|\nabla h(\gamma(\tau))\|^2}{2\sqrt{f(\zeta(\tau)) - f(\zeta(t))}} \\
&\geq \frac{\sigma_{\min}^2(\nabla \Phi^*(\gamma(\tau))) \cdot \|\nabla f(\zeta(\tau))\|^2}{2\sqrt{f(\zeta(\tau)) - f(\zeta(t))}} \\
&\gtrsim \frac{\mu_\Phi^2 \cdot \|\nabla f(\zeta(\tau))\|^2}{\sqrt{f(\zeta(\tau)) - f(\zeta(t))}} \\
&\gtrsim \frac{\sqrt{\alpha_f}\mu_\Phi^2 \cdot \|\nabla f(\zeta(\tau))\|^2}{\|\nabla f(\zeta(\tau))\|} \\
&= \sqrt{\alpha_f}\mu_\Phi^2 \cdot \|\nabla f(\zeta(\tau))\|,
\end{aligned}
\tag{A.8}
$$

provided that the denominators are nonzero. Substituting (A.8) into (A.7), the desired length is bounded by

$$
\begin{aligned}
\ell(t) &\lesssim \nu_\Phi \int_0^t \|\nabla f(\zeta(\tau))\| \, \mathrm{d}\tau \\
&\lesssim -\frac{\nu_\Phi}{\mu_\Phi^2 \sqrt{\alpha_f}} \int_0^t \frac{\mathrm{d}\sqrt{f(\zeta(\tau)) - f(\zeta(t))}}{\mathrm{d}\tau} \, \mathrm{d}\tau \\
&= \frac{\nu_\Phi}{\mu_\Phi^2 \sqrt{\alpha_f}} \left( \sqrt{f(\zeta(0))} - \sqrt{f(\zeta(t))} \right) \\
&\leq \frac{\nu_\Phi \sqrt{f(\zeta(0))}}{\mu_\Phi^2 \sqrt{\alpha_f}} \\
&= \frac{\nu_\Phi \sqrt{h(\gamma(0))}}{\mu_\Phi^2 \sqrt{\alpha_f}} \\
&= \frac{\nu_\Phi \sqrt{h(\mathbf{w}_0)}}{\mu_\Phi^2 \sqrt{\alpha_f}},
\end{aligned}
$$

which completes the proof of Lemma 2.

## C   Proof of Theorem 2

The proof is along the lines of Theorem 1. We first compute the length of the trajectory traversed by gradient descent iterates. Formally, let $I$ denote the first iteration such that $\mathbf{w}_I \notin \mathrm{ball}(\mathbf{w}_0, \rho_\Phi)$. The

length of the trajectory traced by $\{\mathbf{w}_i\}_{i=0}^{I}$ is upper bounded by

$$
\begin{aligned}
\ell(I) &:= \sum_{i=0}^{I-1} \|\mathbf{w}_{i+1} - \mathbf{w}_i\| \\
&= \eta \sum_{i=0}^{I-1} \|\nabla h(\mathbf{w}_i)\| \\
&\lesssim \eta \nu_\Phi \sum_{i=0}^{I-1} \|\nabla f(\mathbf{z}_i)\|.
\end{aligned}
\tag{A.9}
$$

This following lemma is useful for our proof.

**Lemma A.1.** *Suppose* $\mathbf{u}, \mathbf{v} \in \mathrm{ball}(\mathbf{w}_0, \rho_\Phi)$. *Then we have* $\|\Phi(\mathbf{u}) - \Phi(\mathbf{v})\| \leq \frac{3\nu_\Phi}{2}\|\mathbf{u} - \mathbf{v}\|$.

*Proof.* Using Lemma 1, we establish a bound on $\|\Phi(\mathbf{u}) - \Phi(\mathbf{v})\|$:

$$
\begin{aligned}
\|\Phi(\mathbf{u}) - \Phi(\mathbf{v})\| &= \left\| \int_0^1 \nabla\Phi(\mathbf{v} + t(\mathbf{u} - \mathbf{v}))(\mathbf{u} - \mathbf{v}) \, \mathrm{d}t \right\| \\
&\leq \int_0^1 \|\nabla\Phi(\mathbf{v} + t(\mathbf{u} - \mathbf{v}))(\mathbf{u} - \mathbf{v})\| \, \mathrm{d}t \\
&\leq \frac{3\nu_\Phi}{2}\|\mathbf{u} - \mathbf{v}\|.
\end{aligned}
$$

$\square$

Let $i \leq I - 2$. To control the upper bound in (A.9), we use the smoothness of $f$ and Lemma A.1 to obtain a standard "descent inequality" as:

$$
\begin{aligned}
f(\mathbf{z}_i) - f(\mathbf{z}_{i+1}) &\geq \langle \mathbf{z}_i - \mathbf{z}_{i+1}, \nabla f(\mathbf{z}_i) \rangle - \frac{\beta_f}{2}\|\mathbf{z}_{i+1} - \mathbf{z}_i\|^2 \\
&= \langle \Phi(\mathbf{w}_i) - \Phi(\mathbf{w}_{i+1}), \nabla f(\mathbf{z}_i) \rangle - \frac{\beta_f}{2}\|\Phi(\mathbf{w}_{i+1}) - \Phi(\mathbf{w}_i)\|^2 \\
&= \langle \nabla\Phi(\mathbf{w}_i)\{\mathbf{w}_i - \mathbf{w}_{i+1}\}, \nabla f(\mathbf{z}_i) \rangle - \frac{\beta_f}{2}\|\Phi(\mathbf{w}_{i+1}) - \Phi(\mathbf{w}_i)\|^2 \\
&\quad - \langle \Phi(\mathbf{w}_{i+1}) - \Phi(\mathbf{w}_i) - \nabla\Phi(\mathbf{w}_i)\{\mathbf{w}_{i+1} - \mathbf{w}_i\}, \nabla f(\mathbf{z}_i) \rangle \\
&\geq \langle \nabla\Phi(\mathbf{w}_i)\{\mathbf{w}_i - \mathbf{w}_{i+1}\}, \nabla f(\mathbf{z}_i) \rangle - \frac{\beta_f}{2}\|\Phi(\mathbf{w}_{i+1}) - \Phi(\mathbf{w}_i)\|^2 \\
&\quad - \frac{\beta_\Phi}{2}\|\mathbf{w}_{i+1} - \mathbf{w}_i\|^2\|\nabla f(\mathbf{z}_i)\| \\
&\geq \langle \nabla\Phi(\mathbf{w}_i)\{\mathbf{w}_i - \mathbf{w}_{i+1}\}, \nabla f(\mathbf{z}_i) \rangle - \frac{1}{2}\|\mathbf{w}_{i+1} - \mathbf{w}_i\|^2 \left( \beta_\Phi\|\nabla f(\mathbf{z}_i)\| + \frac{9\beta_f\nu_\Phi^2}{4} \right) \\
&= \eta\langle \nabla\Phi(\mathbf{w}_i)\{\nabla h(\mathbf{w}_i)\}, \nabla f(\mathbf{z}_i) \rangle - \frac{\eta^2}{2}\|\nabla h(\mathbf{w}_i)\|^2 \left( \beta_\Phi\|\nabla f(\mathbf{z}_i)\| + \frac{9\beta_f\nu_\Phi^2}{4} \right) \\
&= \eta\|\nabla h(\mathbf{w}_i)\|^2 - \frac{\eta^2}{2}\|\nabla h(\mathbf{w}_i)\|^2 \left( \beta_\Phi\|\nabla f(\mathbf{z}_i)\| + \frac{9\beta_f\nu_\Phi^2}{4} \right) \\
&= \eta\|\nabla h(\mathbf{w}_i)\|^2 \left( 1 - \frac{\eta\beta_\Phi\|\nabla f(\mathbf{z}_i)\|}{2} - \frac{9\eta\beta_f\nu_\Phi^2}{8} \right) \\
&\gtrsim \eta\mu_\Phi^2\|\nabla f(\mathbf{z}_i)\|^2 \quad \text{(chain rule and Lemma 1)}
\end{aligned}
$$

where the fourth inequality holds since $\|\Phi(\mathbf{a}) - \Phi(\mathbf{b}) - \nabla\Phi(\mathbf{b})(\mathbf{a} - \mathbf{b})\| \leq \frac{\beta_\Phi}{2}\|\mathbf{b} - \mathbf{a}\|^2$ for $\beta_\Phi$-smooth $\Phi$, and the last line holds provided that $\eta$ satisfies:

$$
\eta \lesssim \frac{1}{\beta_\Phi \max_i \|\nabla f(\mathbf{z}_i)\| + \beta_f\nu_\Phi^2}.
\tag{A.10}
$$

40    We now use the bound above to find an upper bound on $\sqrt{f(\mathbf{z}_i) - f(\mathbf{z}_{I-1})} - \sqrt{f(\mathbf{z}_{i+1}) - f(\mathbf{z}_{I-1})}$:

$$
\begin{aligned}
\sqrt{f(\mathbf{z}_i) - f(\mathbf{z}_{I-1})} - \sqrt{f(\mathbf{z}_{i+1}) - f(\mathbf{z}_{I-1})} &= \frac{f(\mathbf{z}_i) - f(\mathbf{z}_{i+1})}{\sqrt{f(\mathbf{z}_i) - f(\mathbf{z}_{I-1})} + \sqrt{f(\mathbf{z}_{i+1}) - f(\mathbf{z}_{I-1})}} \\
&\gtrsim \frac{\eta \mu_\Phi^2 \|\nabla f(\mathbf{z}_i)\|^2}{\sqrt{f(\mathbf{z}_i) - f(\mathbf{z}_{I-1})} + \sqrt{f(\mathbf{z}_{i+1}) - f(\mathbf{z}_{I-1})}} \\
&\geq \frac{\eta \mu_\Phi^2 \|\nabla f(\mathbf{z}_i)\|^2}{2\sqrt{f(\mathbf{z}_i) - f(\mathbf{z}_{I-1})}} \\
&\geq \frac{\eta \sqrt{\alpha_f} \mu_\Phi^2 \|\nabla f(\mathbf{z}_i)\|^2}{\sqrt{2}\|\nabla f(\mathbf{z}_i)\|} \\
&= \frac{\eta \sqrt{\alpha_f} \mu_\Phi^2}{\sqrt{2}} \|\nabla f(\mathbf{z}_i)\|.
\end{aligned}
\tag{A.11}
$$

41    Substituting (A.11) into (A.9), we have

$$
\begin{aligned}
\ell(I) &\lesssim \eta \nu_\Phi \sum_{i=0}^{I-1} \|\nabla f(\mathbf{z}_i)\| \\
&\lesssim \frac{\nu_\Phi}{\sqrt{\alpha_f} \mu_\Phi^2} \sum_{i=0}^{I-2} \left( \sqrt{f(\mathbf{z}_i) - f(\mathbf{z}_{I-1})} - \sqrt{f(\mathbf{z}_{i+1}) - f(\mathbf{z}_{I-1})} \right) + \eta \nu_\Phi \|\nabla f(\mathbf{z}_{I-1})\| \\
&\lesssim \frac{\nu_\Phi}{\sqrt{\alpha_f} \mu_\Phi^2} \sqrt{f(\mathbf{z}_0) - f(\mathbf{z}_{I-1})} + \eta \nu_\Phi \|\nabla f(\mathbf{z}_{I-1})\| \\
&\leq \frac{\nu_\Phi \sqrt{f(\mathbf{z}_0)}}{\sqrt{\alpha_f} \mu_\Phi^2} + \eta \nu_\Phi \|\nabla f(\mathbf{z}_{I-1})\|.
\end{aligned}
\tag{A.12}
$$

42    Note that

$$
f(\mathbf{z}_0) = h(\mathbf{w}_0) \lesssim \frac{\alpha_f \mu_\Phi^6}{\beta_\Phi^2 \nu_\Phi^2}
$$

43    and scaling down the learning rate sufficiently to control the second term in the upper bound ensure
44    that

$$
\ell(I) \leq \frac{\rho_\Phi}{2} = \frac{\mu_\Phi}{4\beta_\Phi}.
$$

45    Hence, the gradient descent iterates satisfy:

$$
\{\mathbf{w}_i\}_{i\geq 0} \in \mathrm{ball}(\mathbf{w}_0, \rho_\Phi),
$$

46    which implies that the limit $\overline{\mathbf{w}}$ exists and is globally optimal. In the following, we simplify the
47    expression for $\eta$ in (A.10). Since the iterates of gradient flow remain within a ball of radius $\rho_\Phi$, we
48    can compute the local Lipschitz constant of $f$ as

$$
\begin{aligned}
\max_i \|\nabla f(\mathbf{z}_i)\| &\leq \|\nabla f(\mathbf{z}_0)\| + \max_i \|\nabla f(\mathbf{z}_i) - \nabla f(\mathbf{z}_0)\| \\
&\leq \|\nabla f(\mathbf{z}_0)\| + \beta_f \max_i \|\mathbf{z}_i - \mathbf{z}_0\| \\
&= \|\nabla f(\mathbf{z}_0)\| + \beta_f \max_i \|\Phi(\mathbf{w}_i) - \Phi(\mathbf{w}_0)\| \\
&= \|\nabla f(\mathbf{z}_0)\| + \frac{3\beta_f \nu_\Phi}{2} \max_i \|\mathbf{w}_i - \mathbf{w}_0\| \\
&\leq \|\nabla f(\mathbf{z}_0)\| + \frac{3\beta_f \nu_\Phi}{2} \cdot \rho_\Phi \\
&= \|\nabla f(\mathbf{z}_0)\| + \frac{3\beta_f \mu_\Phi \nu_\Phi}{4\beta_\Phi}.
\end{aligned}
\tag{A.13}
$$

49  Substituting (A.13) into (A.10), an upper bound on $\eta$ is given by

$$\eta \lesssim \frac{1}{\beta_\Phi\|\nabla f(\mathbf{z}_0)\| + \beta_f \mu_\Phi \nu_\Phi + \beta_f \nu_\Phi^2} \leq \frac{1}{\beta_\Phi\|\nabla f(\mathbf{z}_0)\| + \beta_f \mu_\Phi^2 + \beta_f \nu_\Phi^2} \tag{A.14}$$

50  where the last inequality holds since $\mu_\Phi \leq \nu_\Phi$.

51  Finally, using (7), we prove the linear convergence to the limit point $\overline{\mathbf{w}}$:

$$\begin{aligned}
h(\mathbf{w}_{i+1}) &= h(\mathbf{w}_{i+1}) - h(\mathbf{w}_i) + h(\mathbf{w}_i) \\
&= f(\mathbf{z}_{i+1}) - f(\mathbf{z}_i) + h(\mathbf{w}_i) \\
&\leq -C\eta\mu_\Phi^2\|\nabla f(\mathbf{z}_i)\|^2 + h(\mathbf{w}_i) \\
&\leq (1 - C\eta\alpha_f\mu_\Phi^2)h(\mathbf{w}_i)
\end{aligned} \tag{A.15}$$

52  where $C$ is a universal constant. This completes the proof of Theorem 2.

# D    Proof of Lemma 3

54  We first obtain the expression for adjoint operator $\nabla\Phi^*(\Theta) : \mathbb{R}^{d_2 \times n} \to \mathbb{R}^{d_1 \times d_0} \times \mathbb{R}^{d_2 \times d_1}$. Let
55  $\Delta_W \in \mathbb{R}^{d_1 \times d_0}$, $\Delta_V \in \mathbb{R}^{d_2 \times d_1}$, and $\Delta \in \mathbb{R}^{d_2 \times n}$. We expand $\Phi$ as follow:

$$\begin{aligned}
\Phi(W + \Delta_W, V) &\approx \Phi(W, V) + \nabla_W\Phi(\Delta_W), \\
\Phi(W, V + \Delta_V) &\approx \Phi(W, V) + \nabla_V\Phi(\Delta_V)
\end{aligned} \tag{A.16}$$

56  where

$$\nabla_W\Phi(\Delta_W) = V\left(\dot\phi(WX) \odot \Delta_W X\right), \quad \nabla_V\Phi(\Delta_V) = \Delta_V\phi(WX),$$

57  $\odot$ stands for the Hadamard (entry-wise) product, and $\dot\phi(WX)$ is the derivative of $\phi$ calculated at each
58  entry of the matrix $WX$. The operator $\nabla\Phi(\Theta)$ is given by $(\Delta_W, \Delta_V) \to \nabla_W\Phi(\Delta_W) + \nabla_V\Phi(\Delta_V)$.

59  Using the cyclic property of the $\mathrm{trace}$ operator and $\mathrm{trace}\left((A \odot B)C\right) = \mathrm{trace}\left((A \odot C^\top)B^\top\right)$, we
60  have

$$\begin{aligned}
\langle\Delta, \nabla_W\Phi(\Delta_W)\rangle &= \left\langle\left(\dot\phi(WX) \odot V^\top\Delta\right)X^\top, \Delta_W\right\rangle, \\
\langle\Delta, \nabla_V\Phi(\Delta_V)\rangle &= \left\langle\Delta_V, \Delta\phi\left(X^\top W^\top\right)\right\rangle.
\end{aligned} \tag{A.17}$$

61  Substituting (A.17), the adjoint operator is given by

$$\nabla\Phi^*(\Theta) : \Delta \to \left(\left(\dot\phi(WX) \odot V^\top\Delta\right)X^\top, \Delta\phi\left(X^\top W^\top\right)\right). \tag{A.18}$$

62  Suppose that there exist $\dot\phi_{\max}, \ddot\phi_{\max} < \infty$ such that

$$\sup_a |\dot\phi(a)| \leq \dot\phi_{\max}, \quad \sup_a |\ddot\phi(a)| \leq \ddot\phi_{\max}. \tag{A.19}$$

63  **Lemma A.2.** *Let $A \in \mathbb{R}^{m \times n}$ and $B \in \mathbb{R}^{n \times k}$. Then, we have*

$$\sigma_{\min}(A)\|B\| \leq \|AB\| \leq \sigma_{\max}(A)\|B\|.$$

64  Using Lemma A.2 and triangular inequality, we note that

$$\begin{aligned}
\|\nabla\Phi^*(\Theta, \Delta)\| &\leq \left\|\left(\dot\phi(WX) \odot (V^\top\Delta)\right)X^\top\right\| + \left\|\Delta\phi(X^\top W^\top)\right\| \\
&\leq \dot\phi_{\max}\sigma_{\max}(X)\sigma_{\max}(V)\|\Delta\| + \sigma_{\max}(\phi(WX))\|\Delta\|.
\end{aligned} \tag{A.20}$$

65  Similarly, we have this lower bound:

$$\|\nabla\Phi^*(\Theta, \Delta)\| \geq \sigma_{\min}(\phi(WX))\|\Delta\|. \tag{A.21}$$

66 Substituting $\Theta_0 = (W_0, V_0)$ into (A.20) and (A.21), $\mu_\Phi$ and $\nu_\Phi$ are given by:

$$\sigma_{\max}(\nabla\Phi^*(\Theta_0)) \leq \dot{\phi}_{\max}\sigma_{\max}(X)\sigma_{\max}(V_0) + \sigma_{\max}(\phi(W_0 X)) =: \nu_\Phi,$$
$$\sigma_{\min}(\nabla\Phi^*(\Theta_0)) \geq \sigma_{\min}(\phi(W_0 X)) =: \mu_\Phi. \tag{A.22}$$

67 In the following, we find the smoothness parameter $\beta_\Phi$ in (4). Let $\Theta, \hat{\Theta} \in \mathbb{R}^{d_1 \times d_0} \times \mathbb{R}^{d_2 \times d_1}$. We
68 note that $\|\nabla\Phi(\Theta, \Delta) - \nabla\Phi(\hat{\Theta}, \Delta)\| \leq U_1 + U_2$ where

$$U_1 = \|V(\dot{\phi}(W^\top X) \odot (\Delta_W^\top X)) - \hat{V}(\dot{\phi}(\hat{W}^\top X) \odot (\Delta_W^\top X))\|$$
$$U_2 = \|\Delta_V \phi(W^\top X) - \Delta_V \phi(\hat{W}^\top X)\|. \tag{A.23}$$

69 Let us denote

$$\sigma_{\max}(\hat{V}) \leq \chi_{\max}. \tag{A.24}$$

70 An upper bound on $U_1$ in (A.23) is given by:

$$U_1 \leq \|(V - \hat{V})(\dot{\phi}(W^\top X) \odot (\Delta_W^\top X))\| + \|\hat{V}(\dot{\phi}(W^\top X) \odot (\Delta_W^\top X) - \hat{V}\dot{\phi}(\hat{W}^\top X) \odot (\Delta_W^\top X))\|$$
$$\leq \dot{\phi}_{\max}\sigma_{\max}(X)\|V - \hat{V}\|\|\Delta_W\| + \sigma_{\max}(X)\sigma_{\max}(\hat{V})\|\dot{\phi}(W^\top X) - \dot{\phi}(\hat{W}^\top X)\|_\infty\|\Delta_W\|$$
$$\leq \dot{\phi}_{\max}\sigma_{\max}(X)\|V - \hat{V}\|\|\Delta_W\| + \ddot{\phi}_{\max}\sigma_{\max}(X)\|X\|_\infty\sigma_{\max}(\hat{V})\|W - \hat{W}\|\|\Delta_W\|$$
$$\leq \dot{\phi}_{\max}\sigma_{\max}(X)\|V - \hat{V}\|\|\Delta_W\| + \ddot{\phi}_{\max}\chi_{\max}\sigma_{\max}(X)\|W - \hat{W}\|\|\Delta_W\|.$$

71 An upper bound on $U_2$ in (A.23) is given by:

$$U_2 \leq \dot{\phi}_{\max}\sigma_{\max}(X)\|W - \hat{W}\|\|\Delta_V\|.$$

72 Substituting the upper bounds on $U_1$ and $U_2$, an upper bound on $\sigma_{\max}(\nabla\Phi(\Theta) - \nabla\Phi(\hat{\Theta}))$ is given by

$$\sigma_{\max}(\nabla\Phi(\Theta) - \nabla\Phi(\hat{\Theta})) \leq \sigma_{\max}(X)\left(\dot{\phi}_{\max} + \ddot{\phi}_{\max}\chi_{\max}\right)\|W - \hat{W}\| + \sigma_{\max}(X)\dot{\phi}_{\max}\|V - \hat{V}\|$$
$$\leq \sqrt{2}\sigma_{\max}(X)\left(\dot{\phi}_{\max} + \ddot{\phi}_{\max}\chi_{\max}\right)\|\Theta - \hat{\Theta}\|$$

73 where the last inequality holds since

$$\|W - \hat{W}\| + \|V - \hat{V}\| \leq \sqrt{2}\sqrt{\|W - \hat{W}\|^2 + \|V - \hat{V}\|^2}.$$

74 Finally, $\beta_\Phi$ in (4) is given by

$$\beta_\Phi = \sqrt{2}\sigma_{\max}(X)\left(\dot{\phi}_{\max} + \ddot{\phi}_{\max}\chi_{\max}\right). \tag{A.25}$$

## E  Proof of Theorem 3

76 This is our setup: $\min_{\Theta \in R^{d_1 \times d_0} \times R^{d_2 \times d_1}} h(\Theta)$ where

$$h(\Theta) = \|V\phi(WX) - Y\|^2.$$

77 Note that $\alpha_f = \beta_f = 2$.

78 Suppose that there exists $\chi_{\max} < \infty$ such that, for all $i \geq 0$, we have

$$\sigma_{\max}(V_i) \leq \chi_{\max}.$$

79 The details of $\chi_{\max}$ later will be provided in Section E.6.

80 In Lemma 3, we have shown that

$$\mu_\Phi = \sigma_{\min}(\phi(W_0 X)),$$
$$\nu_\Phi = \dot{\phi}_{\max}\sigma_{\max}(X)\sigma_{\max}(V_0) + \sigma_{\max}(\phi(W_0 X)),$$
$$\beta_\Phi = \sqrt{2}\sigma_{\max}(X)\left(\dot{\phi}_{\max} + \ddot{\phi}_{\max}\chi_{\max}\right).$$

81 In order to apply Theorem Theorem 2, we now establish high-probability bounds on random quantities
82 $\mu_\Phi, \nu_\Phi$, and $h(\Theta_0)$ given the initialization in (17).

## E.1  Estimating $\mu_\Phi, \nu_\Phi$

We now estimate the random quantities $\mu_\Phi, \nu_\Phi$ in our neural network setting. They key quantities to estimate are $\sigma_{\min}(\phi(W_0 X))$ and $\sigma_{\max}(\phi(W_0 X))$. To that end, we consider Hermite decomposition of the activation function $\phi$.

We start with the basic definition of Hermite polynomial and its properties. Let $i \geq 0$ and let $q_i : \mathbb{R} \to \mathbb{R}$ denote the $i$-th Hermite polynomial. Note that $q_i$'s form an orthogonal basis for the Hilbert space of functions.:

$$\mathcal{H} = \left\{ u : \mathbb{R} \to \mathbb{R} \mid \int u^2(x) \exp\left(-\frac{x^2}{2}\right) < \infty \right\},$$

which is equipped with the inner product

$$\langle u, v \rangle_{\mathcal{H}} = \frac{1}{\sqrt{2\pi}} \int u(x) v(x) \exp\left(-\frac{x^2}{2}\right) \mathrm{d}x$$

for $u, v \in \mathcal{H}$. We consider probabilist's convention of Hermite polynomial. Specifically, for $i, j \geq 0$, we have

$$\langle q_i, q_j \rangle_{\mathcal{H}} = \begin{cases} i! & i = j, \\ 0 & i \neq j. \end{cases} \tag{A.26}$$

Using the above orthogonal basis to decompose $\phi(W_0 X)$, we have

$$\phi(W_0 X) = \sum_{i=0}^{\infty} \frac{c_i}{i!} \cdot q_i(W_0 X) \tag{A.27}$$

where $c_i = \langle \phi, q_i \rangle_{\mathcal{H}}$ and each matrix $q_i(W_0 X) \in \mathbb{R}^{d_1 \times n}$ is formed by applying $q_i$ entry-wise to the matrix $W_0 X$. Let us denote

$$M_0 := \phi(X^\top W_0^\top) \phi(W_0 X).$$

Let $0 < \tau < 1$. Suppose there are constants $r_1, r_2$ such that $\tau^{r_1} |\phi(a)| \leq |\phi(\tau a)| \leq \tau^{r_2} |\phi(a)|$ for all $a$. In the following, we first obtain $\mathbb{E}[\tilde{M}_0] = \mathbb{E}[\phi(X^\top \tilde{W}_0^\top) \phi(\tilde{W}_0 X)]$ with $\tilde{W}_0 \sim \mathcal{N}(0, 1)$ and then obtain a lower bound on $\sigma_{\min}(\mathbb{E}[M_0])$ and an upper bound on $\sigma_{\min}(\mathbb{E}[M_0])$ by scaling the variance.

Applying Hermite decomposition (A.27) and taking expectation, we have

$$\begin{aligned} \mathbb{E}[\tilde{M}_0] &= \mathbb{E}\left[ \phi(X^\top \tilde{W}_0^\top) \phi(\tilde{W}_0 X) \right] \\ &= \sum_{i,j=0}^{\infty} \frac{c_i c_j}{i! j!} \mathbb{E}[q_i(X^\top \tilde{W}_0^\top) q_j(\tilde{W}_0 X)] \end{aligned} \tag{A.28}$$

where the expectation is w.r.t. the random matrix $\tilde{W}_0$. Let $\mathbf{x}_a \in \mathbb{R}^{d_0}$ denote the $a$-th column of the training data $X$. Each summand in (A.28) is an $n \times n$ matrix where

$$\left[ \mathbb{E}[q_i(X^\top \tilde{W}_0^\top) q_j(\tilde{W}_0 X)] \right]_{a,b} = \sum_{c=1}^{d_1} \mathbb{E}\left[ q_i(\mathbf{x}_a^\top \tilde{W}_{0,c,\rightarrow}) q_j(\tilde{W}_{0,c,\rightarrow}^\top \mathbf{x}_b) \right], \tag{A.29}$$

where $\tilde{W}_{0,c,\rightarrow}$ is the $c$-th row of $\tilde{W}_0$ for $a, b \in [n]$.

In summand on the RHS of (A.29), we note that there is a linear combination of $\tilde{W}_0$'s elements inside of each Hermite polynomial.

We use the properties of Hermite polynomials [3][§18.18.11]:

$$\frac{(a_1^2 + \cdots + a_r^2)^{\frac{i}{2}}}{i!} \tilde{q}_i \left( \frac{a_1 x_1 + \cdots + a_r x_r}{(a_1^2 + \cdots + a_r^2)^{\frac{1}{2}}} \right) = \sum_{s_1 + \cdots + s_r = i} \frac{a_1^{s_1} \cdots a_r^{s_r}}{s_1! \cdots s_r!} \tilde{q}_{s_1}(x_1) \cdots \tilde{q}_{s_r}(x_r) \tag{A.30}$$

where $\tilde{q}_i$'s form an orthogonal basis, equipped with the inner product $\langle u, v \rangle_{\tilde{\mathcal{H}}} = \frac{1}{\sqrt{\pi}} \int u(x) v(x) \exp(-x^2) \mathrm{d}x$. This basis follows the physicist's convention of Hermite polynomial.

108 Since $\tilde{q}_i$ and $q_i$ are rescalings of the other, we can replace $q_i$'s into (A.30). Note that we have
109 $\|\mathbf{x}_a\|_2 = 1$ for all $a \in [n]$. Then we have

$$q_i(\mathbf{x}_a^\top \tilde{W}_{0,c,\to}) = i! \sum_{s_1+\cdots+s_{d_0}=i} \frac{x_{a,1}^{s_1} \cdots x_{a,d_0}^{s_{d_0}}}{s_1! \cdots s_{d_0}!} q_{s_1}(\tilde{W}_{0,c,1}) \cdots q_{s_{d_0}}(\tilde{W}_{0,c,d_0}) \tag{A.31}$$

110 where $x_{a,k}$ and $\tilde{W}_{0,c,k}$ are $k$-th entry of $\mathbf{x}_a$ and $\tilde{W}_{0,c,\to}$ for $k \in [d_0]$. Using the expansion in (A.31),
111 we expand (A.29) as follows:

$$\begin{aligned}
\zeta_{i,j}(a,b) &= i!j! \sum_{s_1+\cdots+s_{d_0}=i} \sum_{s_1'+\cdots+s_{d_0}'=j} \frac{x_{a,1}^{s_1} \cdots x_{a,d_0}^{s_{d_0}}}{s_1! \cdots s_{d_0}!} \cdot \frac{x_{b,1}^{s_1'} \cdots x_{b,d_0}^{s_{d_0}'}}{s_1'! \cdots s_{d_0}'!} \rho_{\mathbf{s},\mathbf{s}'}(\tilde{W}_{0,c,\to}) \\
&= \begin{cases} (i!)^2 \sum_{s_1+\cdots+s_{d_0}=i} \frac{(x_{a,1}x_{b,1})^{s_1} \cdots (x_{a,d_0}x_{b,d_0})^{s_{d_0}}}{s_1! \cdots s_{d_0}!} & i = j, \\ 0 & i \neq j \end{cases} \\
&= \begin{cases} i! \sum_{s_1+\cdots+s_{d_0}=i} \binom{i}{s_1,\cdots,s_{d_0}} (x_{a,1}x_{b,1})^{s_1} \cdots (x_{a,d_0}x_{b,d_0})^{s_{d_0}} & i = j, \\ 0 & i \neq j \end{cases}
\end{aligned} \tag{A.32}$$

112 where $\zeta_{i,j}(a,b) = \mathbb{E}\left[ q_i(\mathbf{x}_a^\top \tilde{W}_{0,c,\to}) q_j(\tilde{W}_{0,c,\to}^\top \mathbf{x}_b) \right]$,

$$\rho_{\mathbf{s},\mathbf{s}'}(\tilde{W}_{0,c,\to}) = \mathbb{E}\left[ q_{s_1}(\tilde{W}_{0,c,1}) \cdots q_{s_{d_0}}(\tilde{W}_{0,c,d_0}) \cdot q_{s_1'}(\tilde{W}_{0,c,1}) \cdots q_{s_{d_0}'}(\tilde{W}_{0,c,d_0}) \right],$$

113 $\mathbf{s} = [s_1, \cdots, s_{d_0}]$, and $\mathbf{s}' = [s_1', \cdots, s_{d_0}']$.

114 To simplify the expression in (A.32), we define $X^{*i} \in \mathbb{R}^{d_0^i \times n}$ where the $a$-th column is given by

$$X_a^{*i} = \mathrm{vec}(\mathbf{x}_a \otimes \cdots \otimes \mathbf{x}_a) \in \mathbb{R}^{d_0^i},$$

115 which is also called Khatri-Rao product. For $i = 0$, we use the convention that $X^{*0} = \mathbf{1}\mathbf{1}^\top \in \mathbb{R}^{n \times n}$.
116 We can rewrite (A.32) as follows:

$$\zeta_{i,j}(a,b) = \begin{cases} i! \langle X_a^{*i}, X_b^{*i} \rangle & i = j \\ 0 & i \neq j. \end{cases} \tag{A.33}$$

117 Substituting (A.33) back into (A.29), we find that

$$\begin{aligned}
\left[ \mathbb{E}[q_i(X^\top \tilde{W}_0^\top) q_j(\tilde{W}_0 X)] \right]_{a,b} &= \sum_{c=1}^{d_1} \mathbb{E}\left[ q_i(\mathbf{x}_a^\top \tilde{W}_{0,c,\to}) q_j(\tilde{W}_{0,c,\to}^\top \mathbf{x}_b) \right] \\
&= \begin{cases} d_1 i! \langle X_a^{*i}, X_b^{*i} \rangle & i = j \\ 0 & i \neq j. \end{cases}
\end{aligned} \tag{A.34}$$

118 Substituting (A.34) into (A.28), we have

$$\mathbb{E}\left[ \tilde{M}_0 \right] = d_1 \left( c_0^2 \mathbf{1}\mathbf{1}^\top + c_1^2 X^\top X + \sum_{i=2}^{\infty} \frac{c_i^2}{i!} (X^{*i})^\top X^{*i} \right). \tag{A.35}$$

119 We now establish an upper bound on $\sigma_{\max}\left( \sum_{i=2}^{\infty} \frac{c_i^2}{i!} (X^{*i})^\top X^{*i} \right)$:

$$\begin{aligned}
\sigma_{\max}\left( \sum_{i=2}^{\infty} \frac{c_i^2}{i!} (X^{*i})^\top X^{*i} \right) &\leq \sum_{i=2}^{\infty} \frac{c_i^2}{i!} \sigma_{\max}((X^{*i})^\top X^{*i}) \\
&\leq c_\infty^2 \sigma_{\max}^2(X)
\end{aligned} \tag{A.36}$$

120 where $c_\infty$ is given by

$$c_\infty^2 = \sum_{i=2}^{\infty} \frac{c_i^2}{i!},$$

121 which is finite provided that $\|\phi\|_{\mathcal{H}}$ is bounded.

122 Using (A.36), we now establish an upper bound on $\sigma_{\max}(\mathbb{E}[\tilde{M}_0])$:

$$\sigma_{\max}(\mathbb{E}[\tilde{M}_0]) \lesssim d_1 \left( nc_0^2 + (c_1^2 + c_\infty^2)\sigma_{\max}^2(X) \right).$$

123 Moreover, suppose there exists some $t$ such that $\sigma_{\min}(X^{*t}) > 0$. This requires to have $d_0^t \geq n$.
124 Putting together the lower bound on $\sigma_{\min}(\mathbb{E}[\tilde{M}_0])$ and the upper bound on $\sigma_{\min}(\mathbb{E}[\tilde{M}_0])$, noting
125 $W_0 = \omega_1 \tilde{W}_0$ and applying $\tau^{r_1}\phi(a) \leq \phi(\tau a) \leq \tau^{r_2}\phi(a)$, we have

$$\omega_1^{2r_1} d_1 \frac{c_t^2}{t!} \sigma_{\min}^2(X^{*t}) \lesssim \sigma_{\min}(\mathbb{E}[M_0]) \leq \sigma_{\max}(\mathbb{E}[M_0]) \lesssim \omega_1^{2r_2} d_1 \left( nc_0^2 + (c_1^2 + c_\infty^2)\sigma_{\max}^2(X) \right).$$
(A.37)

## E.2 Concentration of the random matrix $M_0$

127 To see how well the random matrix $M_0$ concentrates about its expectation, note that

$$
\begin{aligned}
M_0 &= \phi(X^\top W_0^\top)\phi(W_0 X) \\
&= \sum_{i=1}^{d_1} \phi(X^\top W_{0,i,\rightarrow}^\top)\phi(W_{0,i,\rightarrow}X) \\
&= \sum_{i=1}^{d_1} A_i
\end{aligned}
$$
(A.38)

128 where $\{A_i\}_{i=1}^{d_1} \subset \mathbb{R}^{n \times n}$ are independent random matrices.

129 Consider the event $\mathcal{E}_1$ that

$$\max_{i \in [d_1]} \|W_{0,i,\rightarrow}\|_2 \lesssim k_1 \omega_1 \sqrt{d_0 \log d_1}, \quad \max_{i \in [d_1]} \|V_{0,i,\downarrow}\|_2 \lesssim k_2 \omega_2 \sqrt{d_2 \log d_1}$$
(A.39)

130 where $V_{0,i,\downarrow}$ is the $i$-th column of $V_0$. Note that $W_{0,i,\rightarrow} \in \mathbb{R}^{d_0}$ and $V_{0,i,\downarrow} \in \mathbb{R}^{d_2}$ are random zero-
131 mean Gaussian vectors whose entries' variances are $\omega_1^2$ and $\omega_2^2$, respectively. Therefore, with an
132 application of the scalar Bernstein inequality [6, Proposition 5.16], followed by the union bound, we
133 observe that the event $\mathcal{E}_1$ happens except with a probability of at most

$$p_1 := d_1^{-Ck_1 d_0} + d_1^{-Ck_2 d_2},$$
(A.40)

134 for a universal constant $C$ with sufficiently large $k_1, k_2$.

135 Let $i \in [d_1]$. Conditioned on the event $\mathcal{E}_1$, an upper bound on $\|\phi(X^\top W_{0,i,\rightarrow})\|_2$ is given by:

$$\|\phi(X^\top W_{0,i,\rightarrow})\|_2 \lesssim \dot{\phi}_{\max}\sigma_{\max}(X)k_1\omega_1\sqrt{d_0 \log d_1}.$$
(A.41)

136 Moreover, we have

$$
\begin{aligned}
\sigma_{\max}(A_i) &= \|\phi(X^\top W_{0,i,\rightarrow})\|_2^2 \\
&= \|\phi(X^\top W_{0,i,\rightarrow}) - \phi(0)\|_2^2 \\
&\lesssim \dot{\phi}_{\max}^2\sigma_{\max}^2(X)k_1^2\omega_1^2 d_0 \log d_1.
\end{aligned}
$$
(A.42)

137 We now focus on the concentration of $\sigma_{\min}(M_0)$ and $\sigma_{\max}(M_0)$. We use a concentration property,
138 which provides the tail bound of $\tilde{f}(W) = \phi(X^\top W^\top)\phi(WX)$ with multivariate Gaussian input $W$.
139 In the following lemma, we show that $\tilde{f}$ is a Lipschitz function, and its Lipschitz constant explains
140 how $\tilde{f}(W)$ concentrates around its mean.

141 **Lemma A.3.** *Let* $\tilde{f}(W) = \phi(X^\top W^\top)\phi(WX)$. *Suppose* $W$ *satisfies* (A.39). *Then* $\tilde{f}$ *is* $\kappa$-*Lipschitz*
142 *function with constant* $\kappa = 4\dot{\phi}_{\max}^2\sigma_{\max}^2(X)k_1\omega_1\sqrt{d_0 \log d_1}$. *So we have*

$$\|\tilde{f}(W) - \tilde{f}(W')\| < 4\dot{\phi}_{\max}^2\sigma_{\max}^2(X)k_1\omega_1\sqrt{d_0 \log d_1} \cdot \|W - W'\|.$$

143  *Proof.* Note that $\tilde{f}(W_0) = M_0$ and $\tilde{f}$ can be represented as

$$\tilde{f}(X) = \sum_{i=1}^{d_1} f_i(W_{i,\rightarrow})$$

144  where $f_i$ is given by $f_i(W_{i,\rightarrow}) = \phi(X^\top W_{i,\rightarrow}^\top)\phi(W_{i,\rightarrow}X)$. We prove that each $f_i$ is $\kappa$-Lipschitz,
145  which implies that $\tilde{f}$ is also $\kappa$-Lipschitz.

146  We note that $f_i$'s can be expressed as a composition of three functions:

$$f_i(\mathbf{v}) = (g_1 \circ g_2 \circ g_3)(\mathbf{v})$$

147  where $g_1$, $g_2$, and $g_3$ are given by

$$g_1(\mathbf{v}) = \mathbf{v}\mathbf{v}^\top, \ f_2(\mathbf{v}) = \phi(\mathbf{v}), \ f_3(\mathbf{v}) = \mathbf{v}X. \tag{A.43}$$

148  It is clear that $g_2$ is $\dot{\phi}_{\max}$-Lipschitz, and $g_3$ is $\sigma_{\max}(X)$-Lipschitz from their definitions. Lipschitz
149  constant of $g_1$ comes from the domain bound as follows:

$$\begin{aligned}
\|g_1(\mathbf{v} + \delta\mathbf{v}) - g_1(\mathbf{v})\| &= \|\delta\mathbf{v}\mathbf{v}^\top + \mathbf{v}\delta\mathbf{v}^\top + \delta\mathbf{v}\delta\mathbf{v}^\top\| \\
&\leq 2\|\delta\mathbf{v}\mathbf{v}^\top\| + \|\delta\mathbf{v}\delta\mathbf{v}^\top\| \\
&\leq (2\|\mathbf{v}\| + \|\delta\mathbf{v}\|) \cdot \|\delta\mathbf{v}\|.
\end{aligned} \tag{A.44}$$

150  A bound on $(2\|\mathbf{v}\| + \|\delta\mathbf{v}\|)$ is obtained in (A.41). Then $g_1$ is $\kappa_1$-Lipschitz function with
151  $\kappa_1 = 4\dot{\phi}_{\max}\sigma_{\max}(X)k_1\omega_1\sqrt{d_0 \log d_1}$. Therefore, all $g_1$, $g_2$ and $g_3$ are Lipschitz function, so
152  their composition $f_i$ is also Lipschitz function with constant $\kappa = 4\dot{\phi}_{\max}^2\sigma_{\max}^2(X)k_1\omega_1\sqrt{d_0 \log d_1}$,
153  which completes the proof. $\qquad\square$

154  **Lemma A.4.** *Let* $\mathbf{z} \in \mathbb{R}^d$ *denote a Gaussian random vector. Then we have* $\Pr\{\|\mathbf{z} - \mathbb{E}[\mathbf{z}]\| >$
155  $t \,|\mathcal{E}_2\} \lesssim \exp(-t^2)$ *where* $\mathcal{E}_2$ *is the event that* $\|\mathbf{z}\|$ *is bounded.*

156  We can focus on the tail distribution of $M_0 = \tilde{f}(W_0)$. Using Lemmas A.3 and A.4, we have

$$\Pr\{\|M_0 - \mathbb{E}[M_0]\| > t \,|\mathcal{E}_1\} \lesssim \exp(-k_3^2) \tag{A.45}$$

157  where $t = k_3 4\dot{\phi}_{\max}^2\sigma_{\max}^2(X)k_1\omega_1\sqrt{d_0 \log d_1}$ with some constant $k_3$.

158  Using (A.45), we now establish a tail bound on $\sigma_{\min}(M_0)$:

$$\begin{aligned}
\Pr\{\sigma_{\min}(M_0) \leq (1 - \delta_1)\sigma_{\min}(\mathbb{E}[M_0])|\mathcal{E}_1\} &\leq \Pr\{|\sigma_{\min}(M_0) - \sigma_{\min}(\mathbb{E}[M_0])| \geq \delta_1\sigma_{\min}(\mathbb{E}[M_0])|\mathcal{E}_1\} \\
&\leq \Pr\{\sigma_{\min}(M_0 - \mathbb{E}[M_0]) \geq \delta_1\sigma_{\min}(\mathbb{E}[M_0])|\mathcal{E}_1\} \\
&\leq \Pr\{\sigma_{\max}(M_0 - \mathbb{E}[M_0]) \geq \delta_1\sigma_{\min}(\mathbb{E}[M_0])|\mathcal{E}_1\} \\
&\leq \Pr\{\|M_0 - \mathbb{E}[M_0]\| \geq \delta_1\sigma_{\min}(\mathbb{E}[M_0])|\mathcal{E}_1\} \\
&\lesssim p_2
\end{aligned}$$

159  where

$$p_2 = \exp\left(-\left(\frac{\delta_1\sigma_{\min}(\mathbb{E}[M_0])}{4\dot{\phi}_{\max}^2\sigma_{\max}^2(X)k_1\omega_1\sqrt{d_0 \log d_1}}\right)^2\right).$$

160  Similarly, we obtain

$$\Pr\{\sigma_{\max}(M_0) \geq (1 + \delta_2)\sigma_{\max}(\mathbb{E}[M_0])|\mathcal{E}_1\} \lesssim p_3$$

161  where

$$p_3 = \exp\left(-\left(\frac{\delta_2\sigma_{\max}(\mathbb{E}[M_0])}{4\dot{\phi}_{\max}^2\sigma_{\max}^2(X)k_1\omega_1\sqrt{d_0 \log d_1}}\right)^2\right).$$

162  Putting these bounds together with (A.37), we have :

$$\begin{aligned}
\omega_1^{r_1}\sqrt{(1 - \delta_1)\frac{c_t^2}{t!}d_1}\sigma_{\min}(X^{*t}) &\leq \sigma_{\min}(\phi(W_0X)) \\
\sigma_{\max}(\phi(W_0X)) &\leq \sqrt{(1 + \delta_2)}\omega_1^{r_2}(\sqrt{(c_1^2 + c_\infty^2)d_1}\sigma_{\max}(X) + |c_0|\sqrt{d_1n})
\end{aligned} \tag{A.46}$$

163  except with a probability of at most $p_1 + p_2 + p_3$.

164  With establishing the bounds on $\sigma_{\min}(\phi(W_0X))$ and $\sigma_{\max}(\phi(W_0X))$, we can finally estimate $\mu_\Phi, \nu_\Phi$
165  as follows:

### E.3 Lower bound on $\mu_\Phi$

A lower bound on $\mu_\Phi$ is given by

$$\omega_1^{r_1}\sqrt{(1-\delta_1)\frac{c_t^2}{t!}d_1}\,\sigma_{\min}(X^{*t}) \le \sigma_{\min}(\phi(W_0X)) = \mu_\Phi, \tag{A.47}$$

except with a probability of at most $p_1 + p_2$.

### E.4 Upper bound on $\nu_\Phi$

Since $\nu_\Phi = \dot{\phi}_{\max}\sigma_{\max}(X)\sigma_{\max}(V_0) + \sigma_{\max}(\phi(W_0X))$, we obtain a bound on $\sigma_{\max}(V_0)$:

Since $V_0$ is a Gaussian random matrix, we have

$$\sigma_{\max}(V_0) \le \omega_2(2\sqrt{d_1} + \sqrt{d_2}) \lesssim \omega_2\sqrt{d_1} \tag{A.48}$$

except with a probability of at most $p_4 = \exp(-Cd_1)$ where $C$ is a universal constant [6][Corollary 5.35].

Combining (A.48) with the upper bound on $\sigma_{\max}(\phi(W_0X))$, we have

$$\nu_\Phi = \dot{\phi}_{\max}\sigma_{\max}(X)\sigma_{\max}(V_0) + \sigma_{\max}(\phi(W_0X))$$
$$\lesssim \omega_2\dot{\phi}_{\max}\sigma_{\max}(X)\sqrt{d_1} + \omega_1^{r_2}\sqrt{(1+\delta_2)(c_1^2 + c_\infty^2)d_1}\,\sigma_{\max}(X) + \omega_1^{r_2}|c_0|\sqrt{(1+\delta_2)d_1 n}$$

except with a probability of at most $p_1 + p_3 + p_4$.

### E.5 Upper bound on $h(\Theta_0)$

In this section, we bound $h(\Theta_0)$. Using $\|\mathbf{a} + \mathbf{b}\|_2^2 \le 2\|\mathbf{a}\|_2^2 + 2\|\mathbf{b}\|_2^2$, we have

$$\begin{aligned} h(\Theta_0) &= \|V_0\phi(W_0X) - Y\|^2 \\ &\le 2\|V_0\phi(W_0X)\|^2 + 2\|Y\|^2. \end{aligned} \tag{A.49}$$

To upper bound the random norm in (A.49), we first decompose $V_0\phi(W_0X)$ into terms including $W_{0,i,\to} \in \mathbb{R}^{d_0}$ and $V_{0,i,\downarrow} \in \mathbb{R}^{d_2}$ as follows:

$$V_0\phi(W_0X) = \sum_{i=1}^{d_1} B_i \tag{A.50}$$

where $B_i = V_{0,i,\downarrow}\phi(W_{0,i,\to}^\top X) \in \mathbb{R}^{d_2 \times n}$'s are independent random matrices for $i \in [d_1]$.

Conditioned on the event $\mathcal{E}_1$ defined in (A.39), we bound $\|B_i\|$:

$$\begin{aligned} \|B_i\| &= \|V_{0,i,\downarrow}\|_2\|\phi(W_{0,i,\to}^\top X)\|_2 \\ &\le \|V_{0,i,\downarrow}\|_2 \cdot \dot{\phi}_{\max}\sigma_{\max}(X)k_1\omega_1\sqrt{d_0\log d_1} \\ &\le \omega_1\omega_2\dot{\phi}_{\max}\sigma_{\max}(X)k_1k_2\sqrt{d_0d_2}\log d_1 \end{aligned} \tag{A.51}$$

for $i \le d_1$.

Substituting the upper bound in A.50 into A.51 and applying the Hoeffding inequality [2], we have

$$\begin{aligned} \Pr\{\|V_0\phi(W_0X)\| \gtrsim u(d_0,d_1,d_2)|\mathcal{E}_1\} &= \Pr\{\|V_0\phi(W_0X) - \mathbb{E}[V_0\phi(W_0X))|\mathcal{E}_1]\| \gtrsim u(d_0,d_1,d_2)|\mathcal{E}_1\} \\ &\le \Pr\left\{\sum_{i=1}^{d_1}\|B_i - \mathbb{E}[B_i]\| \gtrsim u(d_0,d_1,d_2)|\mathcal{E}_1\right\} \\ &\le p_5 \end{aligned}$$

where

$$u(d_0,d_1,d_2) = \delta_3\omega_1\omega_2\dot{\phi}_{\max}k_1k_2\sqrt{d_0d_1d_2}\,\sigma_{\max}(X)\log d_1$$

185 and $p_5 = \exp(-C\delta_3^2)$ with $\delta_3 \geq 0$ and a universal constant $C$.

186 Therefore, under the event $\mathcal{E}_1$, we have

$$
\begin{aligned}
h(\Theta_0) &\leq 2\|V_0\phi(W_0 X)\|^2 + 2\|Y\|^2 \\
&\lesssim \delta_3^2 \omega_1^2 \omega_2^2 \dot{\phi}_{\max}^2 k_1^2 k_2^2 d_0 d_1 d_2 \sigma_{\max}^2(X) \log^2 d_1 + \|Y\|^2
\end{aligned}
\tag{A.52}
$$

187 except with a probability of at most $p_1 + p_5$. It is natural to assume that $d_2 = o(d_1)$. We also have
188 $\|Y\| \leq 1$.

189 Suppose that

$$
\omega_1 \omega_2 \lesssim \frac{1}{\dot{\phi}_{\max}\sqrt{d_0 d_1}\log d_1}.
\tag{A.53}
$$

190 Substituting (A.53) into (A.52), we have

$$
h(\Theta_0) \leq \delta_3^2 k_1^2 k_2^2 \sigma_{\max}^2(X)
\tag{A.54}
$$

191 where $\delta_3$, $k_1$, and $k_2$ are all constants and independent of $d_0$, $d_1$, and $n$.

## E.6 Denouement

193 The key condition for linear rate convergence of gradient descent in (9) is

$$
h(\Theta_0) \lesssim \frac{\alpha_f \mu_\Phi^6}{\beta_\Phi^2 \nu_\Phi^2}.
$$

194 Putting everything together for the shallow neural network, with high probably, we have

$$
\begin{aligned}
\alpha_f &= 2 \\
\nu_\Phi &= \omega_2 \dot{\phi}_{\max}\sigma_{\max}(X)\sqrt{d_1} + \sqrt{(1+\delta_2)\omega_1^{2r_2}(c_1^2 + c_\infty^2)}\sigma_{\max}(X)\sqrt{d_1} + |c_0|\sqrt{\omega_1^{2r_2}(1+\delta_2)d_1 n} \\
\mu_\Phi &= \omega_1^{r_1}\sqrt{(1-\delta_1)\frac{c_t^2}{t!}d_1}\sigma_{\min}(X^{*t}) \\
\beta_\Phi &= \sqrt{2}\sigma_{\max}(X)\left(\dot{\phi}_{\max} + \ddot{\phi}_{\max}\chi_{\max}\right).
\end{aligned}
\tag{A.55}
$$

195 We note that the order of $\sigma_{\max}(X)$ and $\sigma_{\min}(X^{*t})$ play significant roles for the overparameterization
196 order analysis. For $t = 1$, it requires $n \simeq d_0$, which is not a common setting in practice. In the
197 following, we focus on $t \geq 2$.

## E.7 Order analysis with $t \geq 2$

199 In this section, we assume $|c_0|$ is sufficiently large such that $|c_0|\sqrt{(1+\delta_2)d_1 n}$ becomes the dominat-
200 ing term in $\nu_\Phi$.[1] Then a sufficient condition to satisfy (9) is

$$
d_1^2 \gtrsim \frac{\delta_3^2 c_0^2(1+\delta_2)k_1^2 k_2^2(\dot{\phi}_{\max} + \ddot{\phi}_{\max}\chi_{\max})^2\sigma_{\max}^4(X)nt!^3}{\omega_1^{6r_1-2r_2}(1-\delta_1)^3 c_t^6 \sigma_{\min}^6(X^{*t})},
\tag{A.56}
$$

201 which can be written as

$$
d_1 \gtrsim \sqrt{\frac{\delta_3^2 c_0^2(1+\delta_2)k_1^2 k_2^2(\dot{\phi}_{\max} + \ddot{\phi}_{\max}\chi_{\max})^2 t!^3}{\omega_1^{6r_1-2r_2}(1-\delta_1)^3 c_t^6}} \cdot \frac{\sqrt{n}\sigma_{\max}^2(X)}{\sigma_{\min}^3(X^{*t})}.
$$

202 For notational simplicity, we let $\delta_4 = \max(k_1, k_2)$ and denote $\mathcal{C}_\delta = \{\delta_1, \delta_2, \delta_3, \delta_4\}$ and

$$
\xi(\mathcal{C}_\delta, t, \phi, \{c_i\}_{i \geq 0}) = \sqrt{\frac{\delta_3^2 c_0^2(1+\delta_2)\delta_4^4(\dot{\phi}_{\max} + \ddot{\phi}_{\max}\chi_{\max})^2 t!^3}{\omega_1^{6r_1-2r_2}(1-\delta_1)^3 c_t^6}}.
\tag{A.57}
$$

---

[1]To have a nonzero $c_0$, the activation function should not be an odd function.

203 Note that $\xi(\mathcal{C}_\delta, t, \phi, \{c_i\}_{i\geq 0})$ can be viewed as a constant w.r.t. $d_0$, $d_1$, and $n$. Then (A.56) can be
204 written as:

$$d_1 = \tilde{\Omega}\left(\frac{\sqrt{n}\sigma_{\max}^2(X)}{\sigma_{\min}^3(X^{*t})}\right). \tag{A.58}$$

205 It remains to estimate $\sigma_{\max}(X)$ and $\sigma_{\min}(X^{*t})$ to finish the order analysis of $d_1$. Suppose that $n \simeq d_0^t$.
206 Then , along the lines of [4][Section 2.1], we have $\sigma_{\max}(X) \simeq \sqrt{\frac{n}{d_0}}$ and $\sigma_{\min}(X^{*t}) \simeq \sqrt{\frac{n}{d_0^t}} \simeq 1$.

207 Combining them all, we have

$$d_1 \gtrsim \xi(\mathcal{C}_\delta, t, \phi, \{c_i\}_{i\geq 0})\frac{n^{\frac{3}{2}}}{d_0}. \tag{A.59}$$

208 Therefore, the overall overparameterization degree becomes $d_0 d_1 \simeq \tilde{\Omega}(n^{\frac{3}{2}})$ for $t \geq 2$.

209 The exact expression of $\psi(\phi, \xi, , d_0, d_1, d_2, X)$ in Theorem 3 is given by

$\psi \leq p_1 + p_2 + p_3 + p_4 + p_5$

$\leq d_1^{-C\delta_4 d_0} + d_1^{-C\delta_4 d_2} + e^{-\left(\frac{\delta_1 \sigma_{\min}(\mathbb{E}[M_0])}{4\dot{\phi}_{\max}^2 \sigma_{\max}^2(X)\delta_4\sqrt{d_0 \log d_1}}\right)^2} + e^{-\left(\frac{\delta_2 \sigma_{\max}(\mathbb{E}[M_0])}{4\dot{\phi}_{\max}^2 \sigma_{\max}^2(X)\delta_4\sqrt{d_0 \log d_1}}\right)^2} + e^{-Cd_1} + e^{-C\delta_3^2}.$

210 Note that $d_1^{-C\delta_4 d_0} + d_1^{-C\delta_4 d_2} + \exp(-Cd_1) + \exp(-C\delta_3^2)$ decreases exponentially, which can be
211 sufficiently small without changing the order of $d_1$.

212 Finally, with $d_0 d_1 \simeq \tilde{\Omega}(n^{\frac{3}{2}})$, the gradient descent converges to a global minimum with linear rate
213 with probability at least $1 - \psi$, which can be arbitrary small.

214 **Order analysis without boundedness assumption on $\sigma_{\max}(V_k)$ in Assumption 2.**

215 So far, we assumed $\sigma_{\max}(V_k)$ is bounded for $k \geq 0$. We can relax this assumption by bounding the
216 length of the trajectory of gradient descent as discussed in Appendix C. Recall (A.12):

$$\ell(I) \lesssim \frac{\nu_\Phi \sqrt{f(Z_0)}}{\sqrt{\alpha_f}\mu_\Phi^2}.$$

217 Using triangular inequality and substituting (A.12), we can obtain a bound on $\|V_k\|$

$$\begin{aligned} \|V_k\| &\leq \|V_k - V_0\| + \|V_0\| \\ &\leq \frac{\nu_\Phi \sqrt{f(Z_0)}}{\sqrt{\alpha_f}\mu_\Phi^2} + \|V_0\| \end{aligned} \tag{A.60}$$

218 As shown in (A.48), $\|V_0\| \lesssim \omega_2 \sqrt{d_1}$ with high probability over the choice of $V_0$. With sufficiently
219 small $\omega_2$, the first term in the upper bound dominates in (A.60). Applying (A.54) and substitut-
220 ing (A.60) into (A.56), we have

$$d_1^3 \gtrsim \frac{n^2\sigma_{\max}^6(X)}{\sigma_{\min}^{10}(X^{*t})}$$

$$d_1 \gtrsim \frac{n^{\frac{5}{3}}}{d_0}.$$

221 The overall overparameterization degree becomes $d_0 d_1 \simeq \tilde{\Omega}(n^{\frac{5}{3}})$, which is slightly worse than
222 the result of Theorem 3 under boundedness assumption on $\sigma_{\max}(V_k)$. Note that we still have a
223 subquadratic scaling on the network width.

# F   Additional discussion on lazy training in Section 6

225 In this section, we provide an asymptotic analysis for the term $\|h(\Theta_i) - \tilde{h}(\tilde{\Theta}_i)\|$ to show that there
226 exists a regime where our initialization can avoid lazy training. Recall our setting:

$$\Phi(\Theta) = V \cdot \phi(WX)$$

227 where $W \sim \mathcal{N}(0, \omega_1^2)$ and $V \sim \mathcal{N}(0, \omega_2^2)$. Following the theoretical guidance in (19), we set
228 $\omega_1 \omega_2 \simeq \frac{1}{\sqrt{d_0 d_1}}$.

229 An upper bound on $\|h(\Theta_i) - \tilde{h}(\tilde{\Theta}_i)\|$ is given by [1, Theorem 2.3]:

$$\|h(\Theta_i) - \tilde{h}(\tilde{\Theta}_i)\| \lesssim \frac{\mathrm{Lip}(\nabla\Phi(\Theta))}{\mathrm{Lip}(\Phi(\Theta))^2}. \tag{A.61}$$

230 In the following, we estimate $\frac{\mathrm{Lip}(\nabla\Phi(\Theta))}{\mathrm{Lip}(\Phi(\Theta))^2}$ to find when it is not bound to be close to zero.

231 Substituting $\beta_\Phi$ and $\nu_\Phi$ expressions in (A.55) into the upper bound in (A.61) for sufficiently large
232 $n, c_0$, we have

$$\|h(\Theta_i) - \tilde{h}(\tilde{\Theta}_i)\| \lesssim \frac{\sqrt{2}\sigma_{\max}(X)(\dot{\phi}_{\max} + \ddot{\phi}_{\max}\chi_{\max})}{(\omega_2\dot{\phi}_{\max}\sigma_{\max}(X)\sqrt{d_1} + \omega_1^{r_2}c_0\sqrt{(1+\delta_2)d_1 n})^2}. \tag{A.62}$$

233 We now find an upper bound on $\chi_{\max}$ by bounding the total length of the trajectory of gradient
234 descent as in Appendix C where the length of the trajectory traced by gradient descent is given by
235 (A.12):

$$\ell(I) \le \frac{\nu_\Phi\sqrt{f(Z_0)}}{\sqrt{\alpha_f}\mu_\Phi^2}.$$

236 Using (A.12), (A.48), and (A.54), a bound on $\chi_{\max}$ is given by

$$\begin{aligned}
\|V_i\|_2 &\le \|V_i - V_0\|_F + \|V_0\|_2 \\
&\le \frac{\nu_\Phi\sqrt{f(Z_0)}}{\sqrt{\alpha_f}\mu_\Phi^2} + \|V_0\|_2 \\
&\lesssim \frac{(\omega_2\dot{\phi}_{\max}\sigma_{\max}(X) + \omega_1^{r_2}c_0\sqrt{n})\sigma_{\max}(X)}{\omega_1^{2r_1}\sqrt{d_1}\sigma_{\min}^2(X^{*t})} + \omega_2\sqrt{d_1}
\end{aligned} \tag{A.63}$$

237 Therefore we have

$$\|h(\Theta_i) - \tilde{h}(\tilde{\Theta}_i)\| \lesssim \frac{\sqrt{2}\sigma_{\max}(X)\left(\dot{\phi}_{\max} + \ddot{\phi}_{\max}\frac{(\omega_2\dot{\phi}_{\max}\sigma_{\max}(X)+\omega_1^{r_2}c_0\sqrt{n})\sigma_{\max}(X)}{\omega_1^{2r_1}\sqrt{d_1}\sigma_{\min}^2(X^{*t})} + \omega_2\ddot{\phi}_{\max}\sqrt{d_1}\right)}{(\omega_2\dot{\phi}_{\max}\sigma_{\max}(X)\sqrt{d_1} + \omega_1^{r_2}c_0\sqrt{(1+\delta_2)d_1 n})^2}$$

238 We now consider two cases: 1) $\omega_2\dot{\phi}_{\max}\sigma_{\max}(X) \gtrsim \omega_1^{r_2}c_0\sqrt{n}$ and 2) $\omega_2\dot{\phi}_{\max}\sigma_{\max}(X) \lesssim \omega_1^{r_2}c_0\sqrt{n}$.
239 More precisely, for the asymptomatic analysis, we consider extremal cases $\omega_1 \gg \omega_2$ and $\omega_1 \ll \omega_2$
240 and evaluate $\|h(\Theta_i) - \tilde{h}(\tilde{\Theta}_i)\|$ in each case:

### F.1 Regime with $\omega_2 \gg \omega_1$

242 In the overparameterization regime with large $d$, we note that
243 $\ddot{\phi}_{\max}\frac{(\omega_2\dot{\phi}_{\max}\sigma_{\max}(X)+\omega_1^{r_2}c_0\sqrt{n})\sigma_{\max}(X)}{\omega_1^{2r_1}\sqrt{d_1}\sigma_{\min}^2(X^{*t})} + \omega_2\ddot{\phi}_{\max}\sqrt{d_1} \gtrsim \dot{\phi}_{\max}$. Then we have

$$\|h(\Theta_i) - \tilde{h}(\tilde{\Theta}_i)\| \lesssim \frac{\sqrt{2}\sigma_{\max}(X)\left(\frac{(\omega_2\dot{\phi}_{\max}\sigma_{\max}(X)+\omega_1^{r_2}c_0\sqrt{n})\sigma_{\max}(X)}{\omega_1^{2r_1}\sqrt{d_1}\sigma_{\min}^2(X^{*t})} + \omega_2\sqrt{d_1}\right)}{(\omega_2\dot{\phi}_{\max}\sigma_{\max}(X)\sqrt{d_1} + \omega_1^{r_2}c_0\sqrt{(1+\delta_2)d_1 n})^2}$$

$$\lesssim \frac{\sigma_{\max}^2(X)\left(\frac{\omega_2}{\omega_1^{2r_1}\sqrt{d_1}\sigma_{\min}^2(X^{*t})}\right)}{(\omega_2\sigma_{\max}(X) + \omega_1^{r_2}c_0\sqrt{n})^2 d_1}$$

$$\lesssim \frac{\sigma_{\max}^2(X)\omega_2/d_1^{\frac{3}{2}}}{\sigma_{\min}^2(X^{*t})(\omega_1^{r_1}\omega_2\sigma_{\max}(X) + \omega_1^{r_1+r_2}c_0\sqrt{n})^2}$$

$$\lesssim \frac{\sigma_{\max}^2(X)\omega_2/d_1^{\frac{3}{2}}}{\left(\sigma_{\min}(X^{*t})\sigma_{\max}(X)\frac{\omega_1^{r_1-1}}{\sqrt{d_0 d_1}} + \omega_1^{r_1+r_2}\sigma_{\min}(X^{*t})c_0\sqrt{n}\right)^2}.$$

244 We note that this upper bound above goes to $\infty$ in the regime $\omega_2 \gg \omega_1$, which means that gradient
245 descent can avoid lazy training. Note that it does not imply this training scheme is guaranteed to be
246 non-lazy though.

247 **F.2   Regime with $\omega_1 \gg \omega_2$**

248 In this regime, we have $\ddot{\phi}_{\max}\frac{(\omega_2\dot{\phi}_{\max}\sigma_{\max}(X)+\omega_1^{r_2}c_0\sqrt{n})\sigma_{\max}(X)}{\omega_1^{2r_1}\sqrt{d_1}\sigma_{\min}^2(X^{*t})} \lesssim \dot{\phi}_{\max} + \omega_2\ddot{\phi}_{\max}\sqrt{d_1}$. Then we
249 have

$$\|h(\Theta_i) - \tilde{h}(\tilde{\Theta}_i)\| \lesssim \frac{\sqrt{2}\sigma_{\max}(X)(\dot{\phi}_{\max} + \omega_2\ddot{\phi}_{\max}\sqrt{d_1})}{(\omega_2\dot{\phi}_{\max}\sigma_{\max}(X)\sqrt{d_1} + \omega_1^{r_2}c_0\sqrt{(1+\delta_2)d_1 n})^2}$$

$$\lesssim \frac{\sqrt{2}\sigma_{\max}(X)(\dot{\phi}_{\max} + \omega_2\ddot{\phi}_{\max}\sqrt{d_1})}{(\omega_1^{r_2}c_0\sqrt{d_1 n})^2}. \tag{A.64}$$

250 Note that this bound goes to 0 and lazy training is bound to happen asymptotically.

# G   Implementation details of Section 6

252 For the experiments illustrated in Figure 1, we computed the training and test accuracy for different
253 variants of the proposed weight initialization scheme. We considered the MNIST data set made
254 available through the *torchvision* implementation[2]. We used the provided split of 60 000 training
255 examples and 10 000 test examples which we subsequently normalized.

256 First, a teacher neural network was train on this data set. The label provided by the teacher was then
257 used to relabel both the training and test examples. For each of the weight initializations a student
258 network was constructed and trained on the relabeled data set. The student neural network had 1 000
259 units in its hidden layer and used the GeLU activation function. For the loss we used the mean
260 square error against a one-hot encoding of the true class label. We minimized this loss with stochastic
261 gradient descent (SGD) for which there was three hyperparameter choices. As the difficult of the data
262 set was modest we expected a large range of these hyperparameters to work. It thus sufficed to make
263 a reasonable guess by choosing a batch size of 128, learning rate of 0.01 and 300 epochs. The teacher
264 neural network differed from the student network by using He initialization and cross entropy loss.

265 All results were implemented in PyTorch [5] and run on a Slurm cluster using a Tesla K40c GPU. We
266 fixed $\omega_1\omega_2 \approx 0.002259$ based on the He initialization for our particular network and varied $\omega_2$ in the
267 range $[0.002, 0.1]$. We considered 10 different initialization in this range and ran 5 experiments for
268 each configuration of weight initialization, $(\omega_1, \omega_2)$. Using these independent runs we plotted the
269 mean and standard deviation of the final training and test accuracy in Figure 1, in Section 6.

---

[2]This implementation uses the original MNIST source: http://yann.lecun.com/exdb/mnist/.