# OpenReview forum: "Subquadratic Overparameterization for Shallow Neural Networks"
_NeurIPS.cc/2021/Conference — NeurIPS 2021 Poster_

### Official Review · Reviewer_1Kgg · 2021-07-15

**Rating:** 7
**Confidence:** 4

**Summary:**

This work presents an analysis of the amount of hidden units needed for convergence of a 1 hidden layer neural network to a global minimum under gradient descent (and gradient flow).  In particular, this work establishes a bound of $\tilde{\Omega}(n^{\frac{3}{2}})$ parameters for linear convergence.  Importantly, this work analyzes convergence in a 1 hidden layer network in the case where both weight matrices are trained.  The results also hold for a range of initialization schemes including the Kaiming Normal Initialization.


**Ethical Concerns:**

None at the moment.

**Limitations And Societal Impact:**

The authors discuss some limitations and societal impact appropriately in the conclusion.


**Main Review:**



Positives:
(1) I found the work to be a novel usage of existing techniques for proving convergence in the non-convex setting via the PL inequality.  I found the analysis and the proof idea clear and easy to follow.

(2) The results are novel to the best of my knowledge and the authors provide a nice comparison of how the derived scaling differs with several related works.

(3) Overall, I thought the results were clearly presented.  I especially liked that the authors included upfront a description of the proof technique that would be used throughout the work.  The theoretical results all appear to be correct to the best of my knowledge (I was able to check through Appendix C + G but unfortunately, could only skim through Appendices D-F due to time constraints).

###############################################################

Limitations:
(1) The proof technique is very reminiscent of local convergence results under the PL inequality and smoothness and it might be useful to cite the following related work: https://arxiv.org/abs/2003.00307 , https://arxiv.org/abs/2010.01092.

(2) I think the presentation can be improved slightly by introducing the motivation for analyzing gradient flow first.  Correct me if I’m mistaken, but the proof for the gradient flow case seems to mainly exist to provide intuition for the argument controlling the length of the optimization path in the gradient descent case.  It may be helpful to add text to mention something like this to motivate the gradient flow case a bit more.

(3) This comment is not as related to the main results of the paper, but more to the discussion on the lazy training regime.  I think the authors emphasize quite a bit that the lazy training regime leads to poor generalization, but this is not generally true.  In fact, the infinite width NTK can generalize well in many cases (see https://arxiv.org/abs/1910.01663, https://arxiv.org/abs/2007.15801).  I think the authors may be trying to say that the finite width linearization of a neural network seems to underperform training the same finite width network when all parameters are updated.  I feel this can be clarified a bit throughout the paper (in the introduction & Section 6).

###############################################################

Minor Comments:

(1) I think the function $\psi$ should be real valued for the PL inequality correct? Right now it appears to map to $\mathbb{R}^{d_2}$.  I also think over-parameterization is an implicit assumption for this statement of the PL inequality - this is stated later, but may be useful to say this right before the PL inequality definition.


**Time Spent Reviewing:**

4 hours

---

> ### Author Response · Authors · 2021-08-10
> **Response to Reviewer 1Kgg**
>
> We thank the reviewer for their thoughtful comments and highlighting that "I found the work to be a novel usage of existing techniques for proving convergence in the non-convex setting via the PL inequality. I found the analysis and the proof idea clear and easy to follow. The results are novel to the best of my knowledge and the authors provide a nice comparison of how the derived scaling differs with several related works. Overall, I thought the results were clearly presented. I especially liked that the authors included upfront a description of the proof technique that would be used throughout the work. The theoretical results all appear to be correct to the best of my knowledge".
>
> -----
>
> Concerning the papers that analyze convergence under PL and smoothness conditions, we will cite both of them in the revised version. In [Liu et al., 2021a], the authors establish global convergence when the function to minimize ($h$ in our paper) satisfies a variant of PL condition (local PL condition) assuming the map ($\Phi$ in our paper) is Lipschitz continuous. The setting is different from our paper.
>
> Liu et at., [2021b] characterize  the constancy of the neural tangent kernel via scaling properties of the norm of the Hessian matrix of the network. In this work, we focus on obtaining a sufficient number of parameters for gradient descent to converge to a global minimum with linear rate.
>
>
> [Liu et al., 2021a]: Chaoyue Liu, Libin Zhu, and Mikhail Belkin. Loss landscapes and optimization in over-parameterized non-linear systems and neural networks. In Advances in neural information processing systems (NeurIPS), 2020.
>
> [Liu et al., 2021b]: Chaoyue Liu, Libin Zhu, and Mikhail Belkin. On the linearity of large non-linear models: when and why the tangent kernel is constant. In Advances in neural information processing systems (NeurIPS), 2020.
>
> -------
>
> Concerning the motivation behind considering gradient flow, the intuition is right. In particular, inspired by the analysis of gradient flow, we provide an upper bound on the length of the trajectory traversed by gradient descent iterates and then find the sufficient conditions in terms of initialization to establish convergence to a global minimum. We will comment on this in the revision.
>
> ------
>
> Concerning the generalization ability of the infinite width NTK, in the revision, we will clarify lazy training discussion in Section 6 by citing [Arora et al., 2019] and [Lee et al., 2020] and noting that our experimental results focus on generalization of a network with finite width. For finite width networks, we expect that the original network where all parameters update generalize better than the linearized one.
>
> -------
>
> Regarding minor comments, we will fix the typo in the revision and provide further intuition regarding the PL condition.

---

### Official Review · Reviewer_HcjS · 2021-07-16

**Rating:** 8
**Confidence:** 3

**Summary:**

The paper shows that overparameterized neural networks with $O(n^{3/2})$ neurons, where $n$ is the number of datapoints, are globally convergent at a linear rate with high probability provided that the weights are initialized from Gaussian distributions with the right variances and the activation function is twice-differentiable. The work is the first one to give a result in the regime where the number of neurons is subquadratic in $n$, and unlike previous papers, it considers the case in which both layers are trained.


**Limitations And Societal Impact:**

The authors have addressed the limitations. There is no potential negative societal impact.

**Main Review:**

The argument proposed seems novel and the results obtained are a significative improvement over what was known. The explanation is mostly clear: in particular, Section 3 is useful to get an idea of the proof in the simplified case of gradient flows.

Clarification to be made: In Assumption 2, Line 231, an assumption is made on $sigma_{\max}(V_k)$. Aren’t the matrices $V_k$ the parameters of the optimization problem, i.e. not fixed? How come an assumption is made on them?

Suggestions: When applying Theorem 2 to the case of shallow neural networks, it is mentioned that random matrix theory techniques are applied. Apparently, these techniques are applied in  the proof of Theorem 3, but there is little explanation in the main text about them. It would be good to provide an intuition on them, if space permits. Moreover, the reference to the Khatri-Rao product is hard to parse, given that the concept is not introduced in advance.

**Time Spent Reviewing:**

2.5

---

> ### Author Response · Authors · 2021-08-10
> **Response to Reviewer HcjS**
>
> We thank the reviewer for their thoughtful comments and highlighting that "The argument proposed seems novel and the results obtained are a significative improvement over what was known. The explanation is mostly clear: in particular, Section 3 is useful to get an idea of the proof in the simplified case of gradient flows".
>
> -----
>
> Concerning Assumption 2, we can indeed relax the last assumption. In particular, by bounding the length of the trajectory of gradient descent, using triangular inequality and substituting (A.12 in Appendix C), we establish an upper bound on $||V_k||$ in terms of $f(Z_0)$ and $||V_0||$. With sufficiently small $\omega_2$ and some manipulations, we can show that the overall overparameterization degree becomes $d_0d_1 \simeq \tilde\Omega (n^{\frac{5}{3} })$, which is still subquadratic, but a slightly worse result compare to Theorem 3. In the revision, we will elaborate on this.
>
> -------
>
> To provide further intuition on the techniques used in Theorem 3, the main technique we used is to decompose the random matrix $\phi(X^\top W_0^\top)\phi(W_0 X)$ into independent random matrices and apply various concentration inequalities to establish an upper bound on $\sigma_{\max}(\phi(W_0 X))$ and a lower bound on $\sigma_{\min}(\phi(W_0 X))$ through the Hermite decomposition of $\phi(W_0 X)$. In the revision, we will elaborate on this and define the Khatri-Rao product in the Notations paragraph right after introduction.

---

### Official Review · Reviewer_i9DR · 2021-07-19

**Rating:** 6
**Confidence:** 4

**Summary:**

The focus of this paper is to improve the state-of-the-art overparameterization condition for training shallow neural networks. Besides, this paper also claims that the proposed analytical framework can be applied to the setting that uses standard initialization strategies, which possibly avoids lazy training.

**Limitations And Societal Impact:**

This paper has no negative societal impact.

**Main Review:**

Basically, the main weakness of this paper is regarding the contribution. First, the utilization of smoothness, near-isometry, and PL conditions is not new in the analysis of training overparameterized neural networks. The convergence analysis in [1, 12, 27, 41] has also explicitly or implicitly made use of these properties. Besides, the $O(n^{1/2})$ improvements are not significant and remarkable as this cannot provide us a better understanding of the true learning behavior of GD/SGD for neural network models.

Regarding Table 1, the authors should also clearly state the data assumption made in [34] and this paper in the table. I think the $O(n^2)$ result in [34] is built under the assumption that the training data is Gaussian-like. Besides, [12, 40] are actually considering ReLU activation function while this paper considers smooth activation, which should also be clarified clearly in the table.

The results in Theorem 3 seem to be similar to [34], the authors may need to comment on the difference right after the theorem.

In Section 6, the authors briefly discuss the relationship between initialization to lazy training. However, there might be an issue since the lazy training happens as long as $w_1\cdot w_2$ is large, e.g., $w_1\cdot w_2 = O(1/\sqrt{d_0 d_1})$, instead of the scaling of solely $w_1$ or $w_2$. Different ratio $w_2/w_1$ only determines the learning speed of the first and second layers but does not affect the fact that GD is still practicing lazy training, since $||\Theta_t - \Theta_0||\rightarrow 0$. Therefore, unless considering the regime that $w_1\cdot w_2$ is smaller (e.g, $w_1\cdot w_2<1/d_1$), the convergence of GD will still be in the lazy training regime.

===================

Thank you for providing such a thorough response to my questions. I agree that O(n^{1/2}) improvement is nontrivial and the study of the initialization scaling is also interesting.

I mentioned [1,  27, 41] since I feel that the gradient lower bound results in their paper are similar to the PL condition. So it would be better to comment on this after the definition of PL condition or introduction in your paper.

It would be great if the authors can clarify the difference to [34] since this is the most related work. The results in [34] also have some dependency on the input dimension, it could be better to also comment on the over-parameterization condition proved in [34] when the input dimension is $\Omega(\sqrt{n})$.

The results on the norm between $h(\Theta_i)$ and $\tilde h(\tilde \Theta_i)$ is also interesting. It would also be better to include that lemma in the main part of this paper.

Overall I would like to increase my evaluation to marginally acceptance.

**Time Spent Reviewing:**

3 hrs

---

> ### Author Response · Authors · 2021-08-10
> **Response to Reviewer i9DR**
>
> We thank the reviewer for their thoughtful comments which we address one by one below.
>
> ------
>
> We first address the concern regarding the usage of smoothness, near-isometry, and PL condition in [1, 12, 27, 41].
>
> Allen-Zhu et al. [1] focus on deep networks with ReLU activation assuming data separability, which is different from our setting. In our setting with $L = 2$, the bound in [1] becomes vacuous.
>
> Du et al. [12] assume strict positiveness of the eigenvalues of Gram matrix in a shallow network with ReLU activation. They proved that gradient descent finds a global minimum if the width of the network scales $\tilde\Omega(n^6)$, which is worse than ours.
>
> Li and Liang [27] assume data separability and consider a shallow network with ReLU activation for a classification task, which is a different problem compared to ours.
>
> Zou et al. [41] study the problem of binary classification  assuming data separability for a deep network with ReLU activation, which is a different problem compared to ours.
>
> All these papers consider different settings compared to ours.
>
> ------
>
> Concerning the significance of the $O(n^{1/2})$ improvement, we achieve the best known bounds on the number of parameters by adopting standard initialization strategies and training all layers simultaneously. It is widely accepted that, for two-layer neural networks, the number of parameters should grow linearly with $n$ (e.g., [Kawaguchi and Huang 2019] and [Oymak and Soltanolkotabi, 2020]). Existing results either require much more parameters, or they are established under restrictive settings. In Lines 57-58 and 286-287, we highlighted that "We achieve linear scaling for the width when the number of input features is in $\tilde\Omega(\sqrt{n})$". We believe our results are significant.
>
> Regarding the intuition provided by our paper on better understanding gradient descent, we characterize the length of the trajectory traversed by gradient descent and show that  $\mu_{\Phi}$, $\nu_{\Phi}$, and $\beta_{\Phi}$ play a critical role in the learning behaviour of gradient descent since the required learning rate and initialization for converging to a global minimum are directly determined by those parameters. We then estimate them and establish high-probability bounds for the special case of shallow neural networks.
>
> ------
>
>
> Regarding the suggested improvements to Table 1, these are great points. We will add both data assumption and activation functions for all papers in Table 1 in the revision.
>
> ------
>
> Concerning the relationship with [34], we would like to clarify major differences. Oymak and Soltanolkotabi [34] have shown that $\tilde\Omega(n^2)$ parameters suffice for two-layer neural networks, but only the first layer is trained. Furthermore, even with infinite width, Oymak and Soltanolkotabi [34] cannot guarantee zero training error with probability approaching one. Our results in Theorem 3 are under a more realistic assumption that gradient descent updates $(W, V)$ simultaneously. Furthermore, unlike [34], we adopt standard initialization strategies in Theorem 3. In the revision, we will clarify this after Theorem 3.
>
> ------
>
> Regarding the comment on lazy training, we agree $\omega_1\cdot\omega_2=O(1/\sqrt{d_0d_1})$ seems to fit into an example in [6, Appendix A.2] suggesting lazy training as $d_1\rightarrow \infty$. However, we emphasize that our results are under subquadratic (finite) width. Our initialization is not guaranteed to avoid lazy training. We analyze an upper bound on $\|h(\Theta_i)-\tilde h(\tilde\Theta_i)\|$ using  [6, Theorem 2.3] and show that the upper bound becomes $\infty$ when $\omega_2/\omega_1\rightarrow \infty$. In the revision, we will elaborate on this.

---

### Official Review · Reviewer_N3qa · 2021-07-20

**Rating:** 7
**Confidence:** 3

**Summary:**

In this paper, the author focus on the theoretical framework of minimizing training risk when the loss satisfy PL conditions. They prove the linear convergence in subquadratically over-parameterized shallow networks with smooth activation functions. They conduct experiments on different initialization to support the theory. The theory also provides insights toward avoiding lazy training.

**Limitations And Societal Impact:**

Yes, applying to ReLU is a pretty realistic direction.

**Main Review:**

The paper is well written and smooth to follow. The work looks original to me but I have a few concerns.
1. On experiment perspective, to what extent the theory can even deeper network and relu activation?
2. The experiments are not support the theorem well, no experiments provide to show how width (or overparameterization) relates to convergence and training dynamic.

**Time Spent Reviewing:**

3.5

---

> ### Author Response · Authors · 2021-08-10
> **Response to Reviewer N3qa**
>
> We thank the reviewer for their thoughtful comments and acknowledging that "The paper is well written and smooth to follow. The work looks original to me".
>
> --------
>
> Regarding "On experiment perspective, to what extent the theory can even deeper network and relu activation?":
>
> We would like to highlight that our results in Section 3 and 4 can be leveraged to analyze deeper networks assuming the loss function
> satisfies PL condition. In particular, we provide an upper bound on the length of the trajectory traversed by gradient descent iterates and
> then find the sufficient conditions in terms of initialization to establish convergence to a global minimum. The main challenge we faced to
> extend to deeper networks is to establish an upper bound on $h(\Theta_0)$. In particular, we cannot use a similar technique as in
> Appendix E by decomposing the matrix inside the first norm into a sum of random matrices and then applying concentration inequalities.
>
> Regarding ReLU, we would like to emphasize that our assumptions hold for smooth approximations of ReLU such as GeLU and softplus, which often achieve similar performance in practice (see Remark 2 and associated references). However, to extend to ReLU we would have to deal with isolated non-differentiable points. One potential idea to extend our results to ReLU networks is to define $\nabla\Phi$ using generalized gradients [Frank Clarke, 1975].
>
> Frank H. Clarke. Generalized gradients and applications. Transactions of the American Mathematical Society, 205:247–247, 1975.
>
> --------
>
> Concerning "The experiments do not support the theorem well, no experiments provide to show how width (or overparameterization) relates to convergence and training dynamic":
>
> Note that the starting point for the paper is the well-known experimental observation that the number of parameters only need to exceed the number of training examples by a constant factor. Our goal is to improve the SOTA in terms of the width of the network. Our theory adopts popular He and Lecun initialization that are used widely in practice, for which the effect of overparameterization is widely known experimentally. For detailed experiments, see for example Figure 2 in [Oymak and Soltanolkotabi, 2020], that relates width, dimensionality and the number of samples, and Figure 1 in [Du et al., 2019]. We will make the motivation more precise in the introduction.

---

> > ### Comment · Reviewer_N3qa · 2021-09-01
> > **After author response**
> >
> > Thanks for clarification, I increased my score

---

### Author Response · Authors · 2021-08-10
**Common Response**

We thank Reviewers N3qa, i9DR, HcjS, and 1Kgg for their thoughtful and constructive comments. Reviewer N3qa highlights "The paper is well written and smooth to follow. The work looks original to me". Reviewer HcjS acknowledges that "The argument proposed seems novel and the results obtained are a significant improvement over what was known. The explanation is mostly clear: in particular, Section 3 is useful to get an idea of the proof in the simplified case of gradient flows". Reviewer 1Kgg highlights "I found the work to be a novel usage of existing techniques for proving convergence in the non-convex setting via the PL inequality. I found the analysis and the proof idea clear and easy to follow. The results are novel to the best of my knowledge and the authors provide a nice comparison of how the derived scaling differs with several related works. Overall, I thought the results were clearly presented. I especially liked that the authors included upfront a description of the proof technique that would be used throughout the work. The theoretical results all appear to be correct to the best of my knowledge".

Concerning reviewer i9DR’s comment on the significance of our results, it seems there is a bit of a misunderstanding. We achieve the best known bounds on the number of parameters by adopting standard initialization strategies and training all layers simultaneously. In particular, we note that when the number of features scales as $\sqrt{n}$, we achieve linear scaling for the width as mentioned in Lines 57-58 and 286-287. We elaborate on this in our direct response.

The reviews will further help us improve the final version. To this end, we answer the individual concerns of reviewers in the order that each reviewer listed them. We believe that our response satisfactorily addresses the issues and we look forward to receiving your feedback.

---

### Decision · Program_Chairs · 2021-09-27

**Decision:**

Accept (Poster)

**Comment:**

This paper proves $O(n^{3/2})$ overparameterization condition for training two-layer neural networks when the activation function is smooth and the loss function is square loss. The majority of the reviewers are in support of accepting this paper. Thus, I recommend acceptance. In the camera-ready, the authors should make it clear that the comparison in Table 1 is under different assumptions on the activation functions (e.g., smooth activation functions vs. ReLU).